# Sample-Efficient Preference-based Reinforcement Learning with Dynamics Aware Rewards

**Katherine Metcalf**     **Miguel Sarabia**     **Natalie Mackraz**     **Barry-John Theobald**
Apple, California, USA
{kmetcalf, miguelsdc, natalie_mackraz, barryjohn_theobald}@apple.com

**Abstract:** Preference-based reinforcement learning (PbRL) aligns a robot behavior with human preferences via a reward function learned from binary feedback over agent behaviors. We show that dynamics-aware reward functions improve the sample efficiency of PbRL by an order of magnitude. In our experiments we iterate between: (1) learning a dynamics-aware state-action representation $z^{sa}$ via a self-supervised temporal consistency task, and (2) bootstrapping the preference-based reward function from $z^{sa}$, which results in faster policy learning and better final policy performance. For example, on quadruped-walk, walker-walk, and cheetah-run, with 50 preference labels we achieve the same performance as existing approaches with 500 preference labels, and we recover 83% and 66% of ground truth reward policy performance versus only 38% and 21%. The performance gains demonstrate the benefits of *explicitly* learning a dynamics-aware reward model. Repo: `https://github.com/apple/ml-reed`.

**Keywords:** human-in-the-loop learning, preference-based RL, RLHF

## 1 Introduction

The quality of a reinforcement learned (RL) policy depends on the quality of the reward function used to train it. However, specifying a reliable numerical reward function is challenging. For example, a robot may learn to maximize a defined reward function without actually completing a desired task, known as reward hacking [1, 2]. Instead, preference-based reinforcement learning (PbRL) infers the reward values by way of preference feedback used to train a policy [3, 4, 5, 6, 7, 8, 9, 10, 11, 12]. Using preference feedback avoids the need to manually define absolute numerical reward values (e.g. in TAMER [13]) and is easier to provide than corrective feedback (e.g. in DAGGER [14]). However, many existing PbRL methods require either demonstrations [5], which are not always feasible to provide, or an impractical number of feedback samples [3, 4, 11, 12, 15].

We target sample-efficient reward function learning by exploring the benefits of dynamics-aware preference-learned reward functions or **R**ewards **E**ncoding **E**nvironment **D**ynamics (REED) (Section 4.1). Fast alignment between robot behaviors and human needs is essential for robots operating on real world domains. Given the difficulty people face when providing feedback for a single state-action pair [13], and the importance of defining preferences over transitions instead of single state-action pairs [4], *it is likely that people's internal reward functions are defined over outcomes rather than state-action pairs*. We hypothesize that: (1) modelling the relationship between state, action, and next-state triplets is essential to learn preferences over transitions, (2) encoding awareness of dynamics with a temporal consistency objective will allow the reward function to better generalize over states and actions with similar outcomes, and (3) exposing the reward model to all transitions experienced by the policy during training will result in more stable reward estimations during reward and policy learning. Therefore, we incorporate environment dynamics via a self-supervised temporal consistency task using the state-of-the-art self-predictive representations (SPR) [16] as one such method for capturing environment dynamics.

7th Conference on Robot Learning (CoRL 2023), Atlanta, USA.

We evaluate the benefits of dynamics-awareness using the current state-of-the-art in preference learning [3, 4]. In our experiments, which follow Lee et al. [10], REED reward functions outperform non-REED reward functions across different preference dataset sizes, quality of preference labels, observation modalities, and tasks (Section 5). *REED reward functions lead to faster policy training and reduce the number of preference samples needed (Section 6) supporting our hypotheses about the importance of environments dynamics for preference-learned reward functions.*

## 2 Related Work

**Learning from Human Feedback.** Learning reward functions from preference-based feedback [7, 17, 18, 19, 20, 21, 22, 23] has been used to address the limitations of learning policies directly from human feedback [24, 25, 26] by inferring reward functions from either task success [27, 28, 29] or real-valued reward labels [30, 31]. Learning policies directly from human feedback is inefficient as near constant supervision is commonly assumed. Inferring reward functions from task success feedback requires examples of success, which can be difficult to acquire in complex and multi-step task domains. Finally, people have difficulty providing reliable, real-valued reward labels. PbRL was extended to deep RL domains by Christiano et al. [3], then improved upon and made more efficient by PEBBLE [4] followed by SURF [11], Meta-Reward-Net (MRN) [12], and RUNE [15]. To reduce the feedback complexity of PbRL, PEBBLE [4] sped up policy learning via (1) intrinsically-motivated exploration, and (2) relabelling the experience replay buffer. Both techniques improved the sample complexity of the policy and the trajectories generated by the policy, which were then used to seek feedback. SURF [11] reduced feedback complexity by incorporating augmentations and pseudo-labelling into the reward model learning. RUNE [15] improved feedback sample complexity by guiding policy exploration with reward uncertainty. MRN [12] incorporated policy performance in reward model updates, but further investigation is required to ensure that the method does not allow the policy to influence and bias how the reward function is learned, two concerns called out in [32]. SIRL [33] adds an auxiliary contrastive objective to encourage the reward function to learn similar representations for behaviors human labellers consider to be similar. However, this approach requires extra feedback from human teachers to provide information about which behaviors are similar to one another. Additionally, preference-learning has also been incorporated into data-driven skill extraction and execution in the absence of a known reward function [34]. Of the extensions and improvements to PbRL, only MRN [12] and SIRL [33], like REED, explore the benefits of auxiliary information.

**Encoding Environment Dynamics.** Prior work has demonstrated the benefits of encoding environment dynamics in the state-action representation of a policy [16, 35, 36], and reward functions for imitation learning [37, 38] and inverse reinforcement learning [39, 40]. Additionally, it is common for dynamics models to predict both the next state and the environment's reward [35], which suggests it is important to imbue the reward function with awareness of the dynamics. The primary self-supervised approach to learning a dynamics model is to predict the latent next state [16, 35, 41, 42], and the current state-of-the-art in data efficient RL [16, 36] uses SPR [16] to do exactly this. Unlike prior work in imitation and inverse reinforcement learning, we explicitly evaluate the benefits of *dynamics-aware auxiliary* objectives versus auxiliary objective induced regularization effect.

## 3 Preference-based Reinforcement Learning

RL trains an agent to achieve tasks via environment interactions and reward signals [43]. For each time step $t$ the environment provides a state $s_t$ used by the agent to select an action according to its policy $a_t \sim \pi_\phi(a|s_t)$. Then $a_t$ is applied to the environment, which returns a next state according to its transition function $s_{t+1} \sim \tau(s_t, a_t)$ and a reward $r(s_t, a_t)$. The agent's goal is to learn a policy $\pi_\phi$ maximizing the expected discounted return, $\sum_{k=0}^{\infty} \gamma^k r(s_{t+k}, a_{t+k})$. In PbRL [3, 4, 5, 7, 8, 9, 10] $\pi_\phi$ is trained with a reward function $\hat{r}_\psi$ distilled from preferences $P_\psi$ iteratively queried from a teacher, where $r_\psi$ is assumed to be a latent factor explaining the preference $P_\psi$. A buffer $\mathcal{B}$ of transitions is accumulated as $\pi_\phi$ learns and explores.

A labelled preference dataset $\mathcal{D}_{\text{pref}}$ is acquired by querying a teacher for preference labels every $K$ steps of policy training and is stored as triplets $(\sigma^1, \sigma^2, y_p)$, where $\sigma^1$ and $\sigma^2$ are trajectory segments (sequences of state-action pairs) of length $l$, and $y_p$ is a preference label indicating which, if any, of the trajectories is preferred [10]. To query the teacher, the $M$ *maximally informative* pairs of trajectory segments (e.g. pairs that most reduce model uncertainty) are sampled from $\mathcal{B}$, sent to the teacher for preference labelling, and stored in $\mathcal{D}_{\text{pref}}$ [10, 23, 44, 45]. Typically $\mathcal{D}_{\text{pref}}$ is used to update $\hat{r}_\psi$ on a schedule conditioned on the training steps for $\pi_\phi$ (e.g. every time the teacher is queried).

The preference triplets $(\sigma^1, \sigma^2, y_p)$ create a supervised preference prediction task to approximate $r_\psi$ with $\hat{r}_\psi$ [3, 4, 19]. The prediction task follows the Bradley-Terry model [46] for a stochastic teacher and assumes that the preferred trajectory has a higher cumulative reward according to the teacher's $r_\psi$. The probability of the teacher preferring $\sigma^1$ over $\sigma^2$ ($\sigma^1 \succ \sigma^2$) is formalized as:

$$P_\psi[\sigma^1 \succ \sigma^2] = \frac{\exp \sum_t \hat{r}_\psi(s_t^1, a_t^1)}{\sum_{i \in \{1,2\}} \exp \sum_t \hat{r}_\psi(s_t^i, a_t^i)}, \tag{1}$$

where $s_t^i$ is the state at time step $t$ of trajectory $i \in \{1, 2\}$, and $a_t^i$ is the corresponding action taken.

The parameters $\psi$ of $\hat{r}_\psi$ are optimized such that the binary cross-entropy over $\mathcal{D}_{\text{pref}}$ is minimized:

$$\mathcal{L}^\psi = - \mathop{\mathbb{E}}_{(\sigma^1, \sigma^2, y_p) \sim \mathcal{D}_{\text{pref}}} \big[ y_p(0) \log P_\psi[\sigma^2 \succ \sigma^1] + y_p(1) \log P_\psi[\sigma^1 \succ \sigma^2] \big]. \tag{2}$$

While $P_\psi[\sigma^1 \succ \sigma^2]$ and $\mathcal{L}^\psi$ are defined over trajectory segments, $\hat{r}_\psi$ operates over individual $(s_t, a_t)$ pairs. Each reward estimation in Equation 1 is made *independently* of the other $(s_t, a_t)$ pairs in the trajectory and $P_\psi[\sigma^1 \succ \sigma^2]$ simply sums the independently estimated rewards. Therefore, environment dynamics, or the outcome of different actions in different states, are not explicitly encoded in the reward function, limiting its ability to model the relationship between state-action pairs and the values associated with their outcomes. By supplementing the supervised preference prediction task with a self-supervised temporal consistency task (Section 4.1), we take advantage of all transitions experienced by $\pi_\phi$ to learn a state-action representation in a way that explicitly encodes environment dynamics, and can be used to learn to solve the preference prediction task.

# 4 Dynamics-Aware Reward Function

In this section, we present our approach to encoding dynamics-awareness via a temporal consistency task into the state-action representation of a preference-learned reward function. There are many methods for encoding dynamics and we show results using SPR, the current state of the art. The main idea is to learn a state-action representation that is predictive of the latent representation of the next state using a self-supervised temporal consistency task and all transitions experienced by $\pi_\phi$. Preferences are then learned with a linear layer over the state-action representation.

## 4.1 Rewards Encoding Environment Dynamics (REED)

We use the SPR [16] self-supervised temporal consistency task to learn state-action representations that are predictive of likely future states and thus environment dynamics. The state-action representations are then bootstrapped to solve the preference prediction task in Equation 1 (see Figure 1 for an overview of the architecture). The SPR network is parameterized by $\psi$ and $\theta$, where $\psi$ is shared with $\hat{r}_\psi$ and $\theta$ is unique to the SPR network. At train time, batches of $(s_t, a_t, s_{t+1})$ triplets are sampled from a buffer $\mathcal{B}$ and encoded: $f_s(s_t, \psi_s) \to z_t^s$, $f_a(a_t, \psi_a) \to z_t^a$, $f_{sa}(z_t^s, z_t^a, \psi_{sa}) \to z_t^{sa}$, and $f_s(s_{t+t}, \psi_s) \to z_{t+1}^s$. The embedding $z_{t+1}^s$ is used to form our target for Equations 3 and 4. A dynamics function $g_d(z_t^{sa}, \theta_d) \to \hat{z}_{t+1}^s$ then predicts the latent representation of the next state $z_{t+1}^s$. The functions $f_s(\cdot)$, $f_a(\cdot)$, and $g_d(\cdot)$ are multi-layer perceptrons (MLPs), and $f_{sa}(\cdot)$ concatenates $z_t^s$ and $z_t^a$ along the feature dimension before encoding them with a MLP. To encourage knowledge of environment dynamics in $z_t^{sa}$, $g_d(\cdot)$ is kept simple, e.g. a linear layer.

Following [16], a projection head $h_{\text{pro}}(\cdot, \theta_{\text{pro}})$ is used to project both the predicted and target next state representations to smaller latent spaces via a bottleneck layer and a prediction head

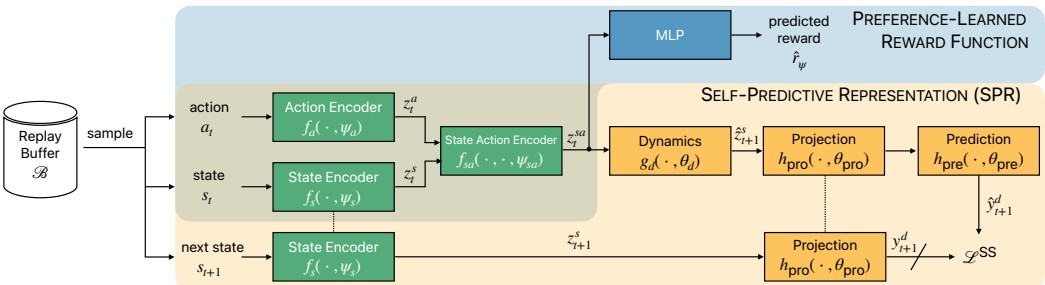

Figure 1: Architecture for self-predictive representation (SPR) objective [16] (in yellow), and preference-learned reward function (in blue). Modules in green are shared between SPR and the preference-learned reward function.

$h_{\text{pre}}(\cdot, \theta_{\text{pre}})$ is used to predict the target projections: $\hat{y}_{t+1}^d = h_{\text{pre}}(h_{\text{pro}}(\hat{z}_{t+1}, \theta_{\text{pro}}), \theta_{\text{pre}})$ and $y_{t+1}^d = h_{\text{pro}}(z_{t+1}, \theta_{\text{pro}})$. Both $h_{\text{pro}}$ and $h_{\text{pre}}$ are modelled using linear layers.

The benefits of REED should be independent of the self-supervised objective function. Therefore, we present results for two different self-supervised objectives: Distillation (i.e. SimSiam with loss $\mathcal{L}^{\text{SS}}$) [36, 47] and Contrastive (i.e. SimCLR with loss $\mathcal{L}^{\text{C}}$) [48, 49, 50], referred to as Distillation REED and Contrastive REED respectively. $\mathcal{L}^{\text{SS}}$ and $\mathcal{L}^{\text{C}}$ are defined as:

$$\mathcal{L}^{\text{SS}} = -\cos(\hat{y}_{t+1}^d, \text{sg}(y_{t+1}^d)), \tag{3}$$

$$\mathcal{L}^{\text{C}} = -\log \frac{\exp(\cos(\hat{y}_{t+1}^d, \text{sg}(y_{t+1}^d))/\tau)}{\sum_{k=1}^{2N} \mathbb{1}_{[s_k \neq s_{t+1}]} \exp(\cos(y_{t+1}^d, \hat{y}_k^d)/\tau)}. \tag{4}$$

In $\mathcal{L}^{\text{SS}}$, a stop gradient operation, sg(...), is applied to $y_{t+1}^d$ and then $\hat{y}_{t+1}^d$ is pushed to be consistent with $y_{t+1}^d$ via a negative cosine similarity loss. In $\mathcal{L}^{\text{C}}$, a stop gradient operation is applied to $y_{t+1}^d$ and then $\hat{y}_{t+1}^d$ is pushed to be predictive of which candidate next state is the true next state via the NT-Xent loss.

Rather than applying augmentations to the input, temporally adjacent states are used to create the different views [16, 36, 49, 50]. Appendix D details the architectures for the SPR component networks.

**State-Action Fusion Reward Network.** REED requires a modification to the reward network architecture used by Christiano et al. [3] and PEBBLE [4] as latent *state* representations are compared. Instead of concatenating raw state-action features, we separately encode the state, $f_s(\cdot)$, and action, $f_a(\cdot)$, before concatenating the embeddings and passing them to the body of our reward network. For the purposes of comparison, we refer to the modified reward network as the state-action fusion (SAF) reward network. For architecture details, see Appendix D.

### 4.2 Incorporating REED into PbRL

The self-supervised temporal consistency task is used to update the parameters $\psi$ and $\theta$ each time the reward network is updated (every $K$ steps of policy training, Section 3). All transitions in the buffer $\mathcal{B}$ are used to update the state-action representation $z^{sa}$, which effectively increases the amount of data used to train the reward function from $M \cdot K$ preference triplets to all state-action pairs experienced by the policy[1]. REED precedes selecting and presenting the $M$ queries to the teacher for feedback. Updating $\psi$ and $\theta$ *prior* to querying the teacher exposes $z^{sa}$ to a larger space of environment dynamics (all transitions collected since the last model update), which enables the model to learn more about the world prior to selecting informative trajectory pairs for the teacher to label. The state-action representation $z^{sa}$ and a linear prediction layer are used to solve the

---

[1]Note the reward function is still trained with $M \cdot K$ triplets, but the state-action encoder has the opportunity to better capture the dynamics of the environment.

preference prediction task (Equation 1). After each update to $\hat{r}_\psi$, $\pi_\phi$ is trained on the updated $\hat{r}_\psi$. See Appendix C for REED incorporated into PrefPPO [3] and PEBBLE [4].

## 5  Experimental Setup

Our experimental results in Section 6 demonstrate that learning a dynamics-aware reward function explicitly improves PbRL policy performance for both state-space and image-space observations. To verify that the performance improvements are due to dynamics-awareness rather than just the inclusion of a self-supervised auxiliary task, we compared against an image-augmentation-based auxiliary task (Image Aug.). The experiments and results are provided in Appendix G) and show that indeed the performance improvements are due specifically to encoding environment dynamics in the reward function. Additionally, we compare against the SURF [11], RUNE [15], and MRN [12] extensions to PEBBLE.

We follow the experiments outlined by the B-Pref benchmark [10]. Models are evaluated on the DeepMind Control Suite (DMC) [51] and MetaWorld [52] environment simulators. DMC provides locomotion tasks with varying degrees of difficulty and MetaWorld provides object manipulation tasks. For each DMC and MetaWorld task, we evaluate performance on varying amounts of feedback, i.e. different preference dataset sizes, and different labelling strategies for the synthetic teacher. The number of queries ($M$) presented to the teacher every $K$ steps is set such that for a given task, teacher feedback always stops at the same episode. Feedback is provided by simulated teachers following [3, 4, 10, 11, 34], where six labelling strategies are used to evaluate model performance under different types and amounts of label noise. The teaching strategies were first proposed by B-Pref [10]. An overview of the labelling strategies is provided in Appendix B.

Following Christiano et al. [3] and PEBBLE [4], $\hat{r}_\psi$ is modelled with an ensemble of three networks with a corresponding ensemble for the SPR auxiliary task. The ensemble is key for disagreement-based query sampling (Appendix A) and has been shown to improve final policy performance [10]. All queried segments are of a fixed length ($l = 50$)[2]. The Adam optimizer [53] with $\beta_1 = 0.9$, $\beta_2 = 0.999$, and no $L_2$-regularization [54] is used to train the reward functions. For all PEBBLE-related methods, intrinsic policy training is reported in the learning curves and occurs over the first 9000 steps. The batch size for training on the preference dataset is $M$, matching the number of queries presented to the teacher, and varies based on the amount of feedback. For details about model architectures, hyper-parameters, and the image augmentations used in the image-augmentation self-supervised auxiliary task, refer to Appendices D, E, and G. None of the hyper-parameters nor architectures are altered from the original SAC [55], PPO [56], PEBBLE [4], PrefPPO [4], Meta-Reward-Net [12], RUNE [15], nor SURF [11] papers. The policy and preference learning implementations provided in the B-Pref repository [57] are used for all experiments.

## 6  Results

The synthetic preference labellers allow policy performance to be evaluated against the ground truth reward function and is reported as mean and standard deviation over 10 runs. Both learning curves and mean normalized returns are reported, where mean normalized returns [10] are given by:

$$\text{normalized returns} = \frac{1}{T} \sum_t \frac{r_\psi(s_t, \pi_\phi^{\hat{r}_\psi}(a_t))}{r_\psi(s_t, \pi_\phi^{r_\psi}(a_t))}, \tag{5}$$

where $T$ is the number of policy training training steps or episodes, $r_\psi$ is the ground truth reward function, $\pi_\phi^{\hat{r}_\psi}$ is the policy trained on the learned reward function, and $\pi_\phi^{r_\psi}$ is the policy trained on the ground truth reward function.

---

[2]Fixed segments lengths are not strictly necessary, and, when evaluating with simulated humans, are harmful when the reward is a constant step penalty.

Table 1: Mean and $\pm$ s.d. normalized return (Equation 5) over 10 random seeds with the oracle labeller and disagreement sampling. The best result for each condition is in **bold**. BASE refers to the PEBBLE or PrefPPO baseline, +DISTILL distillation REED, and +CONTRAST contrastive REED. SURF, RUNE, and MRN are baselines. Results are normalized relative to SAC. See Appendices H.2 and H.4 for all tasks, feedback amounts, and PREFPPO results.

| TASK | FEED. | PEBBLE | | | | | |
| --- | --- | --- | --- | --- | --- | --- | --- |
| | | BASE | +DISTILL | +CONTRAST | SURF [11] | RUNE [15] | MRN [12] |
| WALKER WALK | 500 | $0.74 \pm 0.18$ | $0.86 \pm 0.20$ | $\mathbf{0.90 \pm 0.17}$ | $0.78 \pm 0.12$ | $0.76 \pm 0.20$ | $0.77 \pm 0.20$ |
| | 50 | $0.21 \pm 0.10$ | $\mathbf{0.66 \pm 0.24}$ | $0.62 \pm 0.22$ | $0.47 \pm 0.13$ | $0.23 \pm 0.12$ | $0.38 \pm 0.12$ |
| QUADRUPED WALK | 500 | $0.56 \pm 0.21$ | $\mathbf{1.10 \pm 0.21}$ | $\mathbf{1.10 \pm 0.21}$ | $0.80 \pm 0.18$ | $\mathbf{1.10 \pm 0.20}$ | $\mathbf{1.10 \pm 0.21}$ |
| | 50 | $0.38 \pm 0.26$ | $0.65 \pm 0.16$ | $0.31 \pm 0.18$ | $0.48 \pm 0.19$ | $0.44 \pm 0.21$ | $\mathbf{0.83 \pm 0.12}$ |
| CHEETAH RUN | 500 | $0.86 \pm 0.14$ | $0.88 \pm 0.22$ | $\mathbf{0.94 \pm 0.21}$ | $0.56 \pm 0.16$ | $0.61 \pm 0.17$ | $0.80 \pm 0.16$ |
| | 50 | $0.35 \pm 0.11$ | $0.63 \pm 0.23$ | $\mathbf{0.70 \pm 0.28}$ | $0.55 \pm 0.18$ | $0.32 \pm 0.12$ | $0.38 \pm 0.16$ |
| BUTTON PRESS | 10K | $0.66 \pm 0.26$ | *Collapses* | $0.65 \pm 0.27$ | $\mathbf{0.68 \pm 0.29}$ | $0.45 \pm 0.21$ | $0.59 \pm 0.27$ |
| | 2.5K | $0.37 \pm 0.18$ | *Collapses* | $\mathbf{0.49 \pm 0.25}$ | $0.40 \pm 0.18$ | $0.22 \pm 0.10$ | $0.35 \pm 0.15$ |
| SWEEP INTO | 10K | $0.28 \pm 0.12$ | *Collapses* | $0.47 \pm 0.23$ | $\mathbf{0.48 \pm 0.26}$ | $0.29 \pm 0.15$ | $0.28 \pm 0.25$ |
| | 2.5K | $0.15 \pm 0.09$ | *Collapses* | $0.21 \pm 0.13$ | $\mathbf{0.25 \pm 0.13}$ | $0.16 \pm 0.11$ | $0.22 \pm 0.12$ |
| MEAN | - | 0.46 | 0.47 | **0.64** | 0.55 | 0.46 | 0.57 |

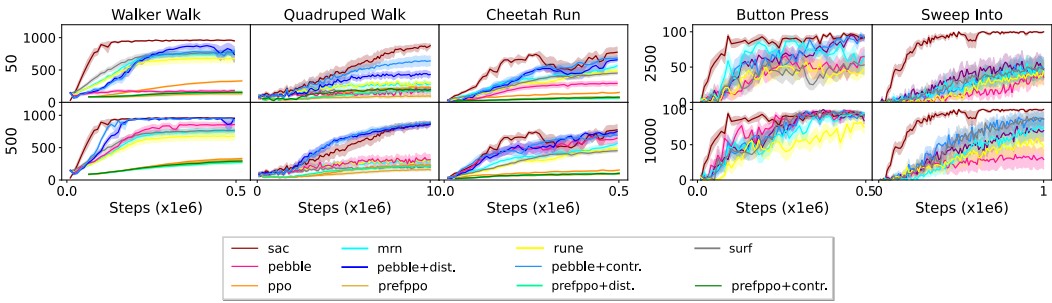

Figure 2: Learning curves for three DMC and two MetaWorld tasks with 50 and 500 (DMC) and 2.5k and 10k (MetaWorld) pieces of feedback, for state-space observations, disagreement sampling, and oracle labels. Refer to Appendices H.1 and H.3 for more tasks and feedback amounts.

Learning curves for state-space observations for SAC and PPO trained on the ground truth reward, PEBBLE, PrefPPO, PEBBLE + REED, PrefPPO + REED, Meta-Reward-Net, SURF, and RUNE are shown in Figure 2, and mean normalized returns [10] are shown in Table 1 (Appendix H.2 and H.4 for PrefPPO). The learning curves show that reward functions learned using REED consistently outperform the baseline methods for locomotive tasks, and for manipulation tasks REED methods are consistently a top performer, especially for smaller amounts of feedback. On average across tasks and feedback amounts, REED methods outperform baselines (MEAN in Table 1).

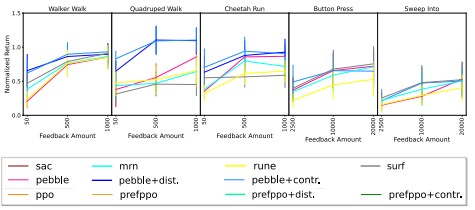

Figure 3: Mean normalized return across oracle, noisy, mistake, and equal labellers Lee et al. [10] on quadruped-walk with state-space observations for 50, 500, and 1000 pieces of feedback.

Learning curves for image-space observations are presented for the PEBBLE and PEBBLE+REED methods in Figure 4, and mean normalized returns [10] in Appendix H.4 . The impact of preference label quality on policy performance for PEBBLE and PEBBLE+REED is shown in Figure 3 and Table 2. On average, across labeller strategies, REED-based methods outperform baselines.

Figures 2, 3 and 4 show that REED improves the speed of policy learning and the final performance of the learned policy relative to non-REED methods. The increase in policy performance is observed across environments, labelling strategies, amount of feedback, and observation type. We ablated the

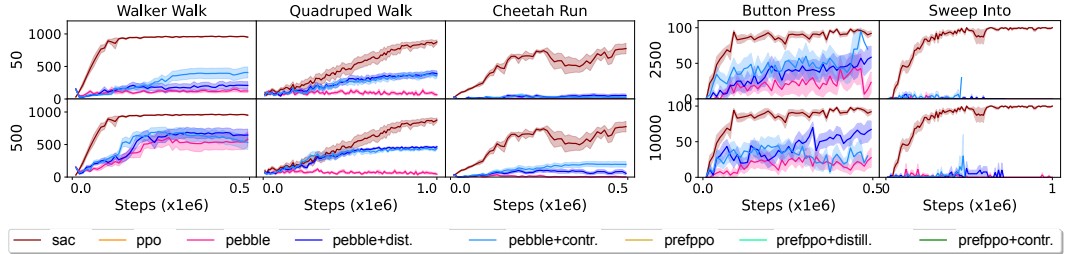

Figure 4: Learning curves for three DMC and two MetaWorld tasks with 50 and 500 (DMC) and 2.5k and 10k (MetaWorld) pieces of feedback, for image–space observations, disagreement sampling, and oracle labels. Only PEBBLE is evaluated for the image-space due to the poor state-space performance of PrefPPO. Results for more tasks and feedback amounts are available in Appendices H.1 and H.3.

impact of the modified reward architecture and found that the REED performance improvements are not due to the modified reward network architecture (Appendix F). Refer to Appendix H for results across all combinations of task, feedback amount, and teacher labelling strategies for state-space observations (H.1 and H.2), and image-space MetaWorld results on more tasks (drawer open, drawer close, window open, and door open) and feedback amounts (H.3 and H.4). The trends observed in the subset of tasks included in Table 1, and Figures 2 and 4 are also observed in the additional tasks and experimental conditions in the Appendices.

## 6.1 Source of Improvements

There is no clear advantage between the Distillation and Contrastive REED objectives on the DMC loco-motion tasks, suggesting the improved policy performance stems from encoding awareness of dynamics rather than any particular self-supervised objective.

Table 2: Mean normalized returns across feedback amounts, tasks, and labeller types (oracle, mistake, noisy, and equal).

| PEBBLE | +DISTILL | +CONTRAST | SURF | RUNE | MRN |
|--------|----------|-----------|------|------|-----|
| 0.53   | 0.47     | **0.68**  | 0.54 | 0.46 | 0.50 |

However in the MetaWorld object manipulation tasks, Distillation REED tends to collapse with Contrastive REED being the more robust method. From comparing SAC, PEBBLE, PEBBLE+REED, and PEBBLE+Image Aug. (Appendix G.3), we see that PEBBLE+Image Aug. improves performance over PEBBLE with large amounts of feedback (e.g. 4.2 times higher mean normalized returns for walker-walk at 1000 pieces of feedback), but does not have a large effect on performance for lower-feedback regimes (e.g. 5.6% mean normalized returns with PEBBLE+Image Aug. versus 5.5% with PEBBLE for walker-walk at 50 pieces of feedback). In contrast, incorporating REED always yields higher performance than both the baseline and PEBBLE+Image Aug. regardless of the amount of feedback. For results analyzing the generalizability and stability of reward function learning when using a dynamics-aware auxiliary objective, see Appendix I.

## 7 Discussion and Limitations

The benefits of dynamics awareness are especially pronounced for labelling types that introduce incorrect labels (i.e. mistake and noisy) (Figure 3 and Appendix H) and smaller amounts of preference feedback. For example, on state-space observation DMC tasks with 50 pieces of feedback, REED methods more closely recover the performance of the policy trained on the ground truth reward recovering 62 – 66% versus 21% on walker-walk, and 65 – 85% versus 38% on quadruped-walk for PEBBLE-based methods (Table 1). Additionally, PEBBLE+REED methods retain policy performance with a factor of 10 fewer pieces of feedback compared to PEBBLE. Likewise, when considering image-space observations, PEBBLE+REED methods trained with 10 times less feedback exceed the performance of base PEBBLE on all DMC tasks. For instance, PEBBLE+Contrastive REED achieves a mean normalized return of 53% with 50 pieces of feedback whereas baseline PEBBLE reaches 36% on the same task with 500 pieces of feedback.

The policy improvements are smaller for REED reward functions on MetaWorld tasks than they are for DMC tasks and are generally smaller for PrefPPO than PEBBLE due to a lack of data diversity in the buffer $\mathcal{B}$ used to train on the temporal consistency task. For PrefPPO lack of data diversity is due to slow learning and for MetaWorld a high similarity between observations. In particular, Distillation REED methods on state-space observations frequently suffer representation collapse and are not reported here. The objective, in this case SimSiam, learns a degenerate solution, where states are encoded by an constant function and actions are ignored due to the source and target views having a near perfect cosine similarity. However, representation collapse is not observed for the image-space observations, and baseline performance is retained with one quarter the amount of feedback when training with PEBBLE+REED methods. If the amount of feedback is kept constant, we notice a 25% to 70% performance improvement over the baseline for all PEBBLE+REED methods in the Button Press task.

The benefits of dynamics awareness can be compared against the benefits of other approaches to improving feedback sample complexity, specifically pseudo-labelling (in SURF) [11], guiding policy exploration with reward uncertainty (in RUNE) [15], and incorporating policy performance into reward updates (in MRN) [12]. REED methods consistently outperform SURF, RUNE, and MRN on the DMC tasks demonstrating the importance of dynamics awareness for locomotion tasks. On the MetaWorld object manipulation tasks, REED frequently outperforms SURF, RUNE, and MRN, especially for smaller amounts of feedback, but the performance gains are smaller than for DMC. Smaller performance gains on MetaWorld relative to other sample efficiency methods is in line with general REED findings for MetaWorld (above) that relate to the slower environment dynamics. However, it is important to call out that all four methods are complementary and can be combined.

Across tasks and feedback amounts, policy performance is higher for rewards that are learned on the state-space observations compared to those learned on image-space observations. There are several tasks, such as cheetah-run and sweep into, for which PEBBLE, and therefore all REED experiments that build on PEBBLE, are not able to learn reward functions that lead to reasonable policy performance when using the image-space observations.

The results demonstrate the benefits and importance of environment dynamics to preference-learned reward functions.

**Limitations** The limitations of REED are: (1) more complex tasks still require a relatively large number of preference labels, (2) extra compute and time are required, (3) Distillation REED can collapse when observations have high similarity, and (4) redundant transitions in the buffer $\mathcal{B}$ from slow policy learning or state spaces with low variability result in over-fitting on the temporal consistency task.

# 8 Conclusion

We have demonstrated the benefits of dynamics awareness in a preference-learned reward for PbRL, **especially when feedback is limited or noisy**. Across experimental conditions, we found REED methods retain the performance of PEBBLE with a 10-fold decrease in feedback. The benefits are observed across tasks, observation modalities, and labeller types. Additionally, we found that, compared to the other PbRL extensions targeting sample efficiency, REED most consistently produced the largest performance gains, especially for smaller amounts of feedback. The resulting sample efficiency is necessary for learning reward functions aligned with user preferences in practical robotic settings.

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

## A    Disagreement Sampling

For all experiments in this paper, disagreement sampling is used to select which trajectory pairs will be presented to the teacher for preference labels. Disagreement-based sampling selects trajectory pairs as follows: (1) $N$ segments are sampled uniformly from the replay buffer; (2) the $M$ pairs of segments with the largest variance in preference prediction across the reward network ensemble are sub-sampled. Disagreement-based sampling is used as it reliably resulted in highest performing policies compared to the other sampling methods discussed in Lee et al. [10].

## B    Labelling Strategies

An overview of the six labelling strategies is provided below, ordered from least to most noisy (see [10] for details and configuration specifics):

1. **oracle** - prefers the trajectory segment with the larger return and equally prefers both segments when their returns are identical

2. **skip** - follows oracle, except randomly selects $10\%$ of the $M$ query pairs to discard from the preference dataset $\mathcal{D}_{pref}$

3. **myopic** - follows oracle, except compares discounted returns ($\gamma = 0.9$) placing more weight on transitions at the end of the trajectory

4. **equal** - follows oracle, except marks trajectory segments as equally preferable when the difference in returns is less than $0.5\%$ of the average ground truth returns observed during the last $K$ policy training steps

5. **mistake** - follows oracle, except randomly selects $10\%$ of the $M$ query pairs and assigns incorrect labels in a structured way (e.g., a preference for segment two becomes a preference for segment one)

6. **noisy** - randomly assigns labels with probability proportional to the relative returns associated with the pair, but labels the segments as equally preferred when they have identical returns

# C  REED Algorithm

The REED task is specified in Algorithm 1 in the context of the PEBBLE preference-learning algorithm. The main components of the PEBBLE algorithm are included, with our modifications identified in the comments. For the original and complete specification of PEBBLE, please see [4] - Algorithm 2.

---

**Algorithm 1** PEBBLE + REED Training Procedure

---

1: **Given:**
2:    $K$ ▷ *teacher feedback frequency*
3:    $M$ ▷ *queries per feedback*
4: **Initializes:**
5:    $Q_\theta$ ▷ *parameters for Q-function*
6:    $\hat{r}_\psi$ ▷ *learned reward function*
7:    $\mathrm{SPR}_{(\psi,\theta)}$ ▷ *self-future consistency ($\psi$ parameters shared with $\hat{r}_\psi$)*
8:    $\mathcal{D}_{\mathrm{pref}} \leftarrow \emptyset$ ▷ *preference dataset*
9:    $\mathcal{D}_{\mathrm{SPR}} \leftarrow \emptyset$ ▷ *SPR dataset*

10: ▷ *unsupervised policy training and exploration*     ◁
11: $\mathcal{B}, \pi_\phi \leftarrow \mathrm{EXPLORE}()$ ▷ *[4] - Algorithm 1*

12: ▷ *joint policy and reward training*     ◁
13: **for** policy train step **do**
14:    **if** step % $K = 0$ **then**
15:       $D_{\mathrm{sfc}} \leftarrow D_{\mathrm{sfc}} \bigcup \mathcal{B}$ ▷ *update SPR dataset*
16:       **for** each SPR gradient step **do**
17:          $\{(s_t, a_t, s_{t+1})\} \sim \mathcal{D}_{\mathrm{sfc}}$ ▷ *sample minibatch*
18:          $\{(\hat{z}^s_{t+1}, z^s_{t+1})\} \leftarrow \mathrm{SFC\_FORWARD}(\{(s_t, a_t, s_{t+1})\})$ ▷ *Section D.2*
19:          **optimize** $\mathcal{L}^{\mathrm{reed}}$ with respect to $\mathrm{SPR}_{(\psi,\theta)}$ ▷ *Equations (3) and (4)*
20:       $\hat{r}_\psi \leftarrow \mathrm{SPR}_\psi$ ▷ *copy shared SPR parameters to reward model*
21:       **update** $\mathcal{D}_{\mathrm{pref}}$, $\hat{r}_\psi$, and $\mathcal{B}$ ▷ *following [4] - Algorithm 2 [lines 9 - 18]*
22:    **update** $\mathcal{B}$, $\pi_\phi$, and $Q_\theta$ ▷ *following [4] - Algorithm 2 [lines 20 - 27]*

---

# D  Architectures

The network architectures are specified in PyTorch 1.13. For architecture hyper-parameters, e.g. hidden size and number of hidden layers, see Appendix E.2

## D.1  Self-Predictive Representations Network

The SPR network is implemented in PyTorch. The architectures for the next_state_projector and consistency_predictor when image observations are used come from [58]. The image encoder architecture comes from [59]. The SPR network is initialized as follows:

```
def build_spr_network(
    self,
    state_size: int,
    state_embed_size: int,
    action_size: int,
    action_embed_size: int,
    hidden_size: int,
    consistency_projection_size: int,
    consistency_comparison_hidden_size: int,
    with_consistency_prediction_head: bool,
    num_layers: int,
    image_observations: bool,
    image_hidden_num_channels: int
):
    """
    The network architecture and build logic to the complete the REED
    self-supervised temporal consistency task based on SPR.

    Args:
        state_size: number of features defining the agent's state space
        state_embed_size: number of dimensions in the state embedding
        action_size: number of features defining the agent's actions
        action_embed_size: number of dimensions in the action embedding
        hidden_size: number of dimensions in the hidden layers of
                     state-action embedding network
        consistency_projection_size: number of units used to compare the
                                      predicted and target latent next
                                      state
        consistency_comparison_hidden_size: number of units in the hidden
                                            layers of the next_state_projector
                                            and the consistency_predictor
        with_consistency_prediction_head: when using the contrastive of
                                          objective the consistency
                                          prediction is not used, but is
                                          when using the distillation
                                          objective
        num_layers: number of hidden layers used to embed the
                    state-action representation
        image_observations: whether image observations are used. If image
                            observations are not used, state-space
                            observations are used
        image_hidden_num_channels: the number of channels to use in the
                                   image encoder's hidden layers
    """
    # build the network that will encode the state features
    if image_observations:
        state_conv_encoder = nn.Sequential(
            nn.Conv2d(
                state_size[0],
                image_hidden_num_channels,
                3,
                stride=1
            ),
```

```python
            nn.ReLU(),
            nn.MaxPool2d(2, 2),

            nn.Conv2d(
                image_hidden_num_channels,
                image_hidden_num_channels,
                3,
                stride=1
            ),
            nn.ReLU(),
            nn.MaxPool2d(2, 2),

            nn.Conv2d(
                image_hidden_num_channels,
                image_hidden_num_channels,
                3,
                stride=1
            ),
            nn.ReLU(),
            nn.MaxPool2d(2, 2)
        )
        conv_out_size = torch.flatten(
                _state_conv_encoder(
                    torch.rand(size=[1] + list(state_size))
                )).size()[0]
        self.state_encoder = nn.Sequential(
            _state_conv_encoder
            nn.Linear(conv_out_size, state_embed_size)
            nn.LeakyReLU(negative_slope=1e-2)
        )
    else:
        self.state_encoder = torch.nn.Sequential(
            torch.nn.Linear(state_size, state_embed_size),
            torch.nn.LeakyReLU(negative_slope=1e-2),
        )

    # build the network that will encode the action features
    self.action_encoder = torch.nn.Sequential(
        torch.nn.Linear(action_size, action_embed_size),
        torch.nn.LeakyReLU(negative_slope=1e-2),
    )

    # build the network that models the relationship between the
    # state and action embeddings
    state_action_encoder = []
    hidden_in_size = action_embed_size + state_embed_size
    for i in range(num_layers):
        state_action_encoder.append(
            torch.nn.Linear(hidden_in_size, hidden_size),
        )
        state_action_encoder.append(
            torch.nn.LeakyReLU(negative_slope=1e-2),
        )
        hidden_in_size = hidden_size
    self.state_action_encoder = torch.nn.Sequential(*state_action_encoder)
    # this is a single dense layer because we want to focus as much of
    # the useful semantic information as possible in the state-action
    # representation
    self.next_state_predictor = torch.nn.Linear(
        hidden_size, state_embed_size
    )

    if image_observations:
        self.next_state_projector = nn.Sequential(
            nn.BatchNorm1d(state_embed_size),
```

```python
            nn.ReLU(inplace=True),
            nn.Linear(
                state_embed_size,
                consistency_comparison_hidden_size
            ),
            nn.ReLU(inplace=True),
            nn.Linear(
                consistency_comparison_hidden_size,
                consistency_projection_size
            ),
            nn.LayerNorm(consistency_projection_size)
        )
    if with_consistency_prediction_head:
        self.consistency_predictor = nn.Sequential(
            nn.ReLU(inplace=True),
            nn.Linear(
                consistency_projection_size,
                consistency_comparison_hidden_size
            ),
            nn.ReLU(inplace=True),
            nn.Linear(
                consistency_comparison_hidden_size,
                consistency_projection_size
            ),
            nn.LayerNorm(consistency_projection_size)
        )
    else:
        predictor = None
else:

    self.next_state_projector = torch.nn.Linear(
        state_embed_size,
        consistency_projection_size
    )
    if with_consistency_prediction_head:
        self.consistency_predictor = nn.Linear(
            consistency_projection_size,
            consistency_projection_size
        )
    else:
        self.consistency_predictor = None
```

A forward pass through the SFC network is as follows:

```python
def spr_forward(self,
                transitions: EnvironmentTransitionBatch,
                with_consistency_prediction_head: bool):
    """
    The logic for a forward pass through the SPR network.
    Args:
      transitions: a batch of environment transitions composed of
                   states, actions, and next states
      with_consistency_prediction_head: when using the contrastive of
                                objective the consistency
                                prediction is not used, but is
                                when using the distillation
                                objective
    Returns:
      predicted embedding of the next state - p in SimSiam paper
      next state embedding (detached from graph) - z in SimSiam paper
      dimensionality: (batch, time step)
    """
    # encode the state, the action, and the state-action pair
    # $s_t \rightarrow z_t^s$
    states_embed = self.state_encoder(transitions.states)
```

```python
# a_t → z_t^a
actions_embed = self.action_encoder(transitions.actions)
# (s_t, a_t) → z_t^{sa}
state_action_embeds = torch.concat(
    [states_embed, actions_embed], dim=-1
)
state_action_embed = self.state_action_encoder(
    state_action_embeds
)

# predict and project the representation of the next state
# z_t^{sa} → ẑ_{t+1}^s
next_state_pred = self.next_state_predictor(state_action_embed)
next_state_pred = self.next_state_projector(next_state_pred)
if with_consistency_prediction_head:
    next_state_pred = self.consistency_predictor(next_state_pred)

# we don't want gradients to back-propagate into the learned
# parameters from anything we do with the next state
with torch.no_grad():
    # s_{t+1} → z_{t+1}^s
    # embed the next state
    next_state_embed = self.state_encoder(transitions.next_states)
    # project the next state embedding into a space where it can be
    # compared with the predicted next state
    projected_next_state_embed = self.next_state_projector(
        next_state_embed
    )

# from the SimSiam paper, this is p and z
return next_state_pred, projected_next_state_embed
```

## D.2 SAF Reward Network

The architecture of the SAF Reward Network is a subset of the SFC network with the addition of a linear to map the state-action representation to predicted rewards. The SFC network is implemented in PyTorch and is initialized following the below build method:

```python
def build_saf_network(
    self,
    state_size: int,
    state_embed_size: int,
    action_size: int,
    action_embed_size: int,
    hidden_size: int,
    num_layers: int,
    final_activation_type: str,
    image_observations: bool,
    image_hidden_num_channels: int
):
    """
    Args:
      state_size: number of features defining the agent's state space
      state_embed_size: number of dimensions in the state embedding
      action_size: number of features defining the agent's actions
      action_embed_size: number of dimensions in the action embedding
      hidden_size: number of dimensions in the hidden layers of
                  state-action embedding network
      num_layers: number of hidden layers used to embed the
                  state-action representation
      final_activation_type: the activation used on the final layer
      image_observations: whether image observations are used. If image
                          observations are not used, state-space
                          observations are used
```

```
        image_hidden_num_channels : the number of channels to use in the
                                     image encoder 's hidden layers
    """
        # build the network that will encode the state features
    if image_observations :
        state_conv_encoder = nn . Sequential (
            nn . Conv2d (
                state_size [0] ,
                image_hidden_num_channels ,
                3 ,
                stride =1
            ) ,
            nn . ReLU () ,
            nn . MaxPool2d (2 , 2) ,

            nn . Conv2d (
                image_hidden_num_channels ,
                image_hidden_num_channels ,
                3 ,
                stride =1
            ) ,
            nn . ReLU () ,
            nn . MaxPool2d (2 , 2) ,

            nn . Conv2d (
                image_hidden_num_channels ,
                image_hidden_num_channels ,
                3 ,
                stride =1
            ) ,
            nn . ReLU () ,
            nn . MaxPool2d (2 , 2)
        )
        conv_out_size = torch . flatten (
                _state_conv_encoder (
                    torch . rand ( size =[1] + list ( state_size ))
                )) . size () [0]
        self . state_encoder = nn . Sequential (
            _state_conv_encoder
            nn . Linear ( conv_out_size , state_embed_size )
            nn . LeakyReLU ( negative_slope =1e -2)
        )
    else :
        self . state_encoder = torch . nn . Sequential (
            torch . nn . Linear ( state_size , state_embed_size ),
            torch . nn . LeakyReLU ( negative_slope =1e -2) ,
        )

    # build the network that will encode the action features
    self . action_encoder = torch . nn . Sequential (
        torch . nn . Linear ( action_size , action_embed_size ),
        torch . nn . LeakyReLU ( negative_slope =1e -2) ,
    )

    # build the network that models the relationship between the
    # state and action embeddings
    state_action_encoder = []
    hidden_in_size = action_embed_size + state_embed_size
    for i in range ( num_layers ):
        state_action_encoder . append (
            torch . nn . Linear ( hidden_in_size , hidden_size ),
        )
        state_action_encoder . append (
            torch . nn . LeakyReLU ( negative_slope =1e -2) ,
        )
```

```python
        hidden_in_size = hidden_size
    self.state_action_encoder = torch.nn.Sequential(*state_action_encoder)

    # build the prediction head and select a final activation
    self.prediction_head = torch.nn.Linear(hidden_size, 1)
    if final_activation_type == "tanh":
        self.final_activation = torch.nn.Tanh()
    elif final_activation_type == "sig":
        self.final_activation = torch.nn.Sigmoid()
    else:
        self.final_activation_type = torch.nn.ReLU()
```

A forward pass through the SAF network is as follows:

```python
def saf_forward(self, transitions: EnvironmentTransitionBatch):
    """
    Args:
        transitions: a batch of environment transitions composed of
                     states, actions, and next states
    Returns:
        predicted embedding of the next state - p in SimSiam paper
        next state embedding (detached from graph) - z in SimSiam paper
        dimensionality: (batch, time step)
    """
    # encode the state, the action, and the state-action pair
    # s_t -> z_t^s
    states_embed = self.state_encoder(transitions.states)
    # a_t -> z_t^a
    actions_embed = self.action_encoder(transitions.actions)
    # (s_t, a_t) -> z_t^{sa}
    state_action_embeds = torch.concat(
        [states_embed, actions_embed], dim=-1
    )
    state_action_embed = self.state_action_encoder(
        state_action_embeds
    )

    return self.final_activation(
        self.prediction_head(state_action_embed)
    )
```

# E    Hyper-parameters

## E.1    Train Hyper-parameters

This section specifies the hyper-parameters (e.g. learning rate, batch size, etc) used for the experiments and results (Section 6). The SAC, PPO, PEBBLE, and PrefPPO experiments all match those used in [55], [56], and [4] respectively. The SAC and PPO hyper-parameters are specified in Table 3, the PEBBLE and PrefPPO hyper-parameters are given in Table 4, and the hyper-parameters used to train on the REED task are in Table 5.

The image-space models were trained on images of size 50x50. For PEBBLE and REED on DMC tasks, color images were used, and for MetaWorld tasks, grayscale images were used. All pixel values were scaled to the range $[0.0, 1.0]$.

For the image-based REED methods, we found that a larger value of $k$ was important for the Meta-World experiments compared to the DMC experiments due to slower environment dynamics. In MetaWorld the differences between subsequent observations are far more similar than in DMC. In the state-space, the mean cosine similarity between all observations accumulated in the replay buffer was $0.9$. For the image-space observations, the mean cosine similarity was $0.7$. Additionally, for sweep into, due to the similarity in MetaWorld observations, the slower environment dynamics, and

difficulty of tasks like sweep into, we found it beneficial to update on the REED objective every $5^{\text{th}}$ update to the reward model in order to avoid over fitting on the REED objective and reducing the accuracy of the $\hat{r}_\psi$ preference predictions.

Table 3: Training hyper-parameters for SAC [55] and PPO [56].

| HYPER-PARAMETER | VALUE |
|---|---|
| **SAC** | |
| Learning rate | 1e-3 (cheetah), 5e-4 (walker), 1e-4 (quadruped), 3e-4 (MetaWorld) |
| Batch size | 512 (DMC), 1024 (MetaWorld) |
| Total timesteps | 500k, 1M (quadruped, sweep into) |
| Optimizer | Adam [53] |
| Critic EMA $\tau$ | 5e-3 |
| Critic target update freq. | 2 |
| $(\mathcal{B}_1, \mathcal{B}_2)$ | $(0.9, 0.999)$ |
| Initial Temperature | 0.1 |
| Discount $\gamma$ | 0.99 |
| **PPO** | |
| Learning rate | 5e-5 (DMC), 3e-4 (MetaWorld) |
| Batch size | 128 (all but cheetah), 512 (cheetah) |
| Total timesteps | 500k (cheetah, walker, button press), 1M (quadruped, sweep into) |
| Envs per worker | 8 (sweep into), 16 (cheetah, quadruped), 32 (walker, sweep into) |
| Optimizer | Adam [53] |
| Discount $\gamma$ | 0.99 |
| Clip range | 0.2 |
| Entropy bonus | 0.0 |
| GAE parameter $\lambda$ | 0.92 |
| Timesteps per rollout | 250 (MetaWorld), 500 (DMC) |

Table 4: Training hyper-parameters for PEBBLE [4] and PrefPPO [3, 4]. The only hyper-parameter that differs between PEBBLE and PrefPPO is the DMC learning rate. The batch size for the reward network changes based per total feedback amount to match the number of queries $M$ sent to the teacher for labelling each feedback session.

| HYPER-PARAMETER | VALUE |
|---|---|
| Learning rate PEBBLE | 3e-4 |
| Learning rate PrefPPO | 5e-4 (DMC), 3e-4 (MetaWorld) |
| Optimizer | Adam [53] |
| Segment length $l$ | 50 (DMC), 25 (MetaWorld) |
| Feedback amount / number queries ($M$) | 1k/100, 500/50, 200/20, 100/10, 50/5 (DMC) 20k/100, 10k/50, 5k/25, 2.5k/12 (MetaWorld) |
| Steps between queries ($K$) | 20k (walker, cheetah), 30k (quadruped), 5k (MetaWorld) |

Table 5: Training hyper-parameters for REED with the SPR objective [16] (Section 4.1). The REED hyper-parameters were used with both the PEBBLE [4] and PrefPPO [3, 4] preference-learning algorithms. Hyper-parameters are by environment/task and shared by the two SSL objectives: Distillation versus Contrastive (Section 4.1). Training on the REED task occurred every $K$ steps (specified in Table 4) prior to updating on the preference task. The SPR objective predicts future latent states $k$ steps in the future. While our hyper-parameter sweep evaluated multiple values for $k$, we found that $k = 1$ vs. $k > 1$ had no real impact on learning quality for these state-action feature spaces. For Contrastive REED experiments (state-space and image-space observations), $\tau = 0.005$. In general, the image-based experiments REED were less sensitive to the hyper-parameters than the state-space experiments experiments.

| ENVIRONMENT | LEARNING RATE | EPOCHS PER UPDATE | BATCH SIZE | OPTIMIZER | K |
|---|---|---|---|---|---|
| STATE-SPACE OBSERVATIONS | | | | | |
| Walker | 1e-3 | 20 | 12 | SGD | 1 |
| Cheetah | 1e-3 | 20 | 12 | SGD | 1 |
| Quadruped | 1e-4 | 20 | 128 | Adam [53] | 1 |
| Button Press | 1e-4 | 10 | 128 | Adam [53] | 1 |
| Sweep Into | 5e-5 | 5 | 256 | Adam [53] | 1 |
| IMAGE-SPACE OBSERVATIONS | | | | | |
| Walker | 1e-4 | 5 | 256 | Adam [53] | 1 |
| Cheetah | 1e-4 | 5 | 256 | Adam [53] | 1 |
| Quadruped | 1e-4 | 5 | 256 | Adam [53] | 1 |
| Button Press | 1e-4 | 5 | 256 | Adam [53] | 5 |
| Sweep Into | 1e-4 | 5 | 512 | Adam [53] | 5 |
| Drawer Open | 1e-4 | 5 | 256 | Adam [53] | 5 |
| Drawer Close | 1e-4 | 5 | 256 | Adam [53] | 5 |
| Window Open | 1e-4 | 5 | 256 | Adam [53] | 5 |
| Door Open | 1e-4 | 5 | 256 | Adam [53] | 5 |

## E.2 Architecture Hyper-parameters

The network hyper-parameters (e.g. hidden dimension, number of hidden layers, etc) used for the experiments and results (Section 6) are specified in Table 6.

Table 6: Architecture hyper-parameters for SAC [55], PPO [56], the base reward model (used for PEBBLE [4] and PrePPO [3, 4]), the SAF reward model (Section 4.1), and the SPR model (Section 4.1). The hyper-parameters reported here are intended to inform the values to used to initialize the architectures in Appendix D. Hyper-parameters not relevant to a model are indicated with "N/A". The SPR model is what REED uses to construct the self-supervised temporal consistency task. The base reward model is used with PEBBLE and PrefPPO in Lee et al. [4] and [10]. The SAF reward network is used for all REED conditions in Section 6. The "Final Activation" refers to the activation function used just prior to predicting the reward for a given state action pair. The action embedding sizes are the same for the state-space and image-space observations.

| HYPER-PARAMETER | SAC | PPO | BASE REWARD | SAF REWARD | SPR NET |
|---|---|---|---|---|---|
| STATE-SPACE OBSERVATIONS | | | | | |
| State embed size | N/A | N/A | N/A | 20 (walker), 17 (cheetah), 78 (quadruped), 30 (MetaWorld) | 20 (walker), 17 (cheetah), 78 (quadruped), 30 (MetaWorld) |
| Action embed size | N/A | N/A | N/A | 10 (walker), 6 (cheetah), 12 (quadruped), 4 (MetaWorld) | 10 (walker), 6 (cheetah), 12 (quadruped), 4 (MetaWorld) |
| Comparison units | N/A | N/A | N/A | N/A | 5 (walker), 4 (cheetah), 10 (quadruped), 5 (MetaWorld) |
| Num. hidden | 2 (DMC), 3 (MetaWorld) | 3 | 3 | 3 | 3 |
| Units per layer | 1024 (DMC), 256 (MetaWorld) | 256 | 256 | 256 | 256 |
| Final activation | N/A | N/A | tanh | tanh | N/A |
| IMAGE-SPACE OBSERVATIONS | | | | | |
| State embed size | N/A | N/A | N/A | 20 (walker), 17 (cheetah), 78 (quadruped), 30 (MetaWorld) | 20 (walker), 17 (cheetah), 78 (quadruped), 30 (MetaWorld) |
| Comparison units | N/A | N/A | N/A | N/A | 128 |
| Num. hidden | 2 (DMC), 3 (MetaWorld) | 3 | 3 | 3 | 3 |
| Units per layer | 1024 (DMC), 256 (MetaWorld) | 256 | 256 | 256 | 256 |
| Final activation | N/A | N/A | tanh | tanh | N/A |

# F  SAF Reward Net Ablation

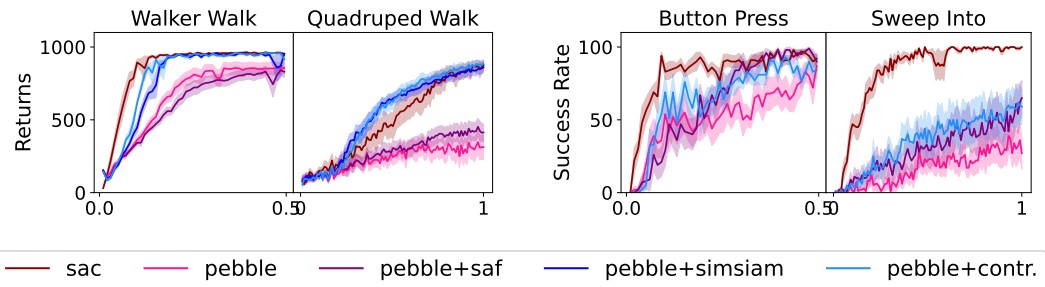

Figure 5: Ablation of the SAF reward net for walker-walk, quadruped-walk, sweep into, and button press with 500 (walker and quadruped) and 5k (sweep into and button press) teacher-labelled queries with disagreement-based sampling and the oracle labelling strategy.

We present results ablating the impact of our modified SAF reward network architecture in Table 7, see Section 4.1, State-Action Fusion Reward Network, for details. In our ablation, we replace the original PEBBLE reward network architecture from [4] with our SAF network and then evaluate on the joint experimental condition with no other changes to reward function learning. We evaluate the impact of the SAF reward network on the walker-walk, quadruped-walk, sweep into, and button press tasks. Policy and reward function learning is evaluated across feedback amounts and labelling styles. All hyper-parameters match those used in all other experiments in the paper (see Appendix E). We compare PEBBLE with the SAF reward network architecture (PEBBLE + SAF) against SAC trained on the ground truth reward, PEBBLE with the original architecture (PEBBLE), PEBBLE with Distillation REED (PEBBLE+Dist.), and PEBBLE with Contrastive REED (PEBBLE+Contr.).

The inclusion of the SAF reward network architecture does not meaningfully impact policy performance. In general, across domains and experimental conditions, PEBBLE + SAF performs on par with or slightly worse than PEBBLE. The lack of performance improvements suggest that the performance improvements observed when the auxiliary temporal consistency objective are due to the auxiliary objective and not the change in network architecture.

Table 7: The impact of the SAF reward network is ablated. Ratio of policy performance on learned versus ground truth rewards for **walker-walk**, **quadruped-walk**, **sweep into**, and **button press** across preference learning methods, labelling methods and feedback amounts (with disagreement sampling).

| FEEDBACK | METHOD | ORACLE | MISTAKE | EQUAL | SKIP | MYOPIC | NOISY | MEAN |
|---|---|---|---|---|---|---|---|---|
| | | | | WALKER-WALK | | | | |
| 1K | PEBBLE | 0.85 (0.17) | 0.76 (0.21) | 0.88 (0.16) | 0.85 (0.17) | 0.79 (0.18) | 0.81 (0.18) | 0.83 |
| | +SAF | 0.81 (0.19) | 0.62 (0.18) | 0.88 (0.16) | 0.81 (0.19) | 0.74 (0.17) | 0.81 (0.19) | 0.78 |
| | +DIST. | 0.9 (0.16) | 0.77 (0.2) | 0.91 (0.12) | 0.89 (0.16) | 0.8 (0.17) | 0.88 (0.17) | 0.86 |
| | +CONTR. | 0.9 (0.16) | 0.77 (0.2) | 0.91 (0.12) | 0.89 (0.16) | 0.8 (0.17) | 0.88 (0.17) | 0.86 |
| 500 | PEBBLE | 0.74 (0.18) | 0.61 (0.17) | 0.84 (0.19) | 0.75 (0.19) | 0.67 (0.19) | 0.69 (0.19) | 0.72 |
| | +SAF | 0.68 (0.17) | 0.51 (0.13) | 0.76 (0.17) | 0.68 (0.17) | 0.56 (0.15) | 0.68 (0.17) | 0.65 |
| | +DIST. | 0.86 (0.2) | 0.71 (0.2) | 0.87 (0.2) | 0.87 (0.2) | 0.82 (0.22) | 0.84 (0.2) | 0.83 |
| | +CONTR. | 0.9 (0.17) | 0.81 (0.19) | 0.9 (0.14) | 0.9 (0.17) | 0.88 (0.16) | 0.88 (0.18) | 0.88 |
| 250 | PEBBLE | 0.59 (0.17) | 0.41 (0.12) | 0.67 (0.2) | 0.56 (0.17) | 0.43 (0.13) | 0.51 (0.13) | 0.53 |
| | +SAF | 0.53 (0.16) | 0.41 (0.15) | 0.59 (0.18) | 0.53 (0.16) | 0.36 (0.1) | 0.48 (0.14) | 0.48 |
| | +DIST. | 0.8 (0.23) | 0.6 (0.16) | 0.85 (0.21) | 0.8 (0.24) | 0.75 (0.26) | 0.8 (0.24) | 0.77 |
| | +CONTR. | 0.85 (0.19) | 0.73 (0.23) | 0.85 (0.19) | 0.85 (0.2) | 0.79 (0.2) | 0.85 (0.22) | 0.82 |
| | | | | QUADRUPED-WALK | | | | |
| 2K | PEBBLE | 0.94 (0.15) | 0.55 (0.19) | 1.1 (0.26) | 1.0 (0.16) | 0.93 (0.13) | 0.56 (0.19) | 0.86 |
| | +SAF | 0.97 (0.15) | 0.45 (0.17) | 1.2 (0.22) | 0.87 (0.19) | 0.76 (0.13) | 0.59 (0.14) | 0.81 |
| | +DIST. | 1.3 (0.31) | 0.47 (0.19) | 1.4 (0.37) | 1.3 (0.26) | 1.2 (0.18) | 0.96 (0.15) | 1.09 |
| | +CONTR. | 1.3 (0.25) | 0.7 (0.16) | 1.2 (0.24) | 1.3 (0.29) | 1.3 (0.28) | 1.0 (0.16) | 1.13 |
| 1K | PEBBLE | 0.86 (0.15) | 0.53 (0.19) | 0.88 (0.15) | 0.91 (0.14) | 0.73 (0.18) | 0.48 (0.25) | 0.73 |
| | +SAF | 0.79 (0.16) | 0.44 (0.19) | 0.99 (0.23) | 0.9 (0.19) | 0.63 (0.15) | 0.6 (0.2) | 0.72 |
| | +DIST. | 1.1 (0.19) | 0.59 (0.14) | 1.2 (0.22) | 1.3 (0.3) | 1.1 (0.21) | 1.0 (0.15) | 1.04 |
| | +CONTR. | 1.1 (0.19) | 0.63 (0.16) | 1.2 (0.29) | 1.1 (0.19) | 1.1 (0.19) | 0.83 (0.14) | 0.99 |
| 500 | PEBBLE | 0.56 (0.21) | 0.48 (0.21) | 0.66 (0.2) | 0.64 (0.15) | 0.47 (0.22) | 0.48 (0.23) | 0.55 |
| | +SAF | 0.63 (0.16) | 0.4 (0.22) | 0.85 (0.14) | 0.75 (0.19) | 0.56 (0.18) | 0.5 (0.19) | 0.61 |
| | +DIST. | 1.1 (0.21) | 0.58 (0.16) | 1.2 (0.24) | 1.0 (0.22) | 1.0 (0.19) | 0.68 (0.16) | 0.93 |
| | +CONTR. | 1.1 (0.21) | 0.64 (0.11) | 1.1 (0.22) | 1.1 (0.17) | 1.0 (0.17) | 0.85 (0.14) | 0.97 |
| 250 | PEBBLE | 0.53 (0.18) | 0.36 (0.23) | 0.64 (0.15) | 0.62 (0.16) | 0.46 (0.22) | 0.47 (0.21) | 0.51 |
| | +SAF | 0.51 (0.2) | 0.36 (0.22) | 0.73 (0.18) | 0.53 (0.17) | 0.53 (0.19) | 0.45 (0.24) | 0.52 |
| | +DIST. | 0.98 (0.15) | 0.58 (0.18) | 1.0 (0.19) | 0.79 (0.12) | 0.9 (0.18) | 0.77 (0.16) | 0.84 |
| | +CONTR. | 0.98 (0.15) | 0.58 (0.18) | 1.0 (0.19) | 0.79 (0.12) | 0.9 (0.18) | 0.77 (0.16) | 0.84 |
| | | | | BUTTON PRESS | | | | |
| 20K | PEBBLE | 0.72 (0.26) | 0.57 (0.26) | 0.77 (0.25) | 0.75 (0.26) | 0.68 (0.21) | 0.72 (0.24) | 0.70 |
| | +SAF | 0.77 (0.23) | 0.72 (0.28) | 0.84 (0.23) | 0.75 (0.24) | 0.78 (0.21) | 0.77 (0.22) | 0.77 |
| | +CONTR. | 0.65 (0.25) | 0.61 (0.28) | 0.67 (0.27) | 0.67 (0.27) | 0.67 (0.24) | 0.69 (0.26) | 0.66 |
| 10K | PEBBLE | 0.66 (0.26) | 0.47 (0.21) | 0.67 (0.27) | 0.63 (0.26) | 0.67 (0.24) | 0.6 (0.26) | 0.62 |
| | +SAF | 0.7 (0.25) | 0.66 (0.26) | 0.74 (0.23) | 0.71 (0.25) | 0.67 (0.19) | 0.71 (0.25) | 0.70 |
| | +CONTR. | 0.65 (0.27) | 0.61 (0.3) | 0.66 (0.27) | 0.62 (0.26) | 0.6 (0.25) | 0.68 (0.28) | 0.64 |
| 5K | PEBBLE | 0.48 (0.21) | 0.31 (0.12) | 0.56 (0.25) | 0.54 (0.24) | 0.59 (0.23) | 0.52 (0.23) | 0.50 |
| | +SAF | 0.63 (0.25) | 0.55 (0.24) | 0.65 (0.26) | 0.68 (0.24) | 0.62 (0.21) | 0.7 (0.24) | 0.64 |
| | +CONTR. | 0.55 (0.24) | 0.54 (0.26) | 0.65 (0.27) | 0.63 (0.26) | 0.57 (0.24) | 0.63 (0.28) | 0.60 |
| 2.5K | PEBBLE | 0.37 (0.18) | 0.21 (0.088) | 0.44 (0.21) | 0.34 (0.15) | 0.4 (0.17) | 0.34 (0.18) | 0.35 |
| | +SAF | 0.58 (0.26) | 0.38 (0.17) | 0.61 (0.26) | 0.54 (0.23) | 0.52 (0.21) | 0.54 (0.2) | 0.53 |
| | +CONTR. | 0.49 (0.25) | 0.42 (0.22) | 0.52 (0.24) | 0.5 (0.23) | 0.44 (0.17) | 0.45 (0.21) | 0.47 |
| | | | | SWEEP INTO | | | | |
| 20K | PEBBLE | 0.53 (0.25) | 0.26 (0.15) | 0.51 (0.23) | 0.52 (0.27) | 0.47 (0.28) | 0.47 (0.26) | 0.46 |
| | +SAF | 0.5 (0.24) | 0.36 (0.15) | 0.47 (0.22) | 0.39 (0.19) | 0.49 (0.21) | 0.6 (0.21) | 0.47 |
| | +CONTR. | 0.5 (0.22) | 0.36 (0.13) | 0.41 (0.2) | 0.6 (0.22) | 0.54 (0.21) | 0.61 (0.25) | 0.50 |
| 10K | PEBBLE | 0.28 (0.12) | 0.22 (0.13) | 0.45 (0.21) | 0.33 (0.17) | 0.47 (0.25) | 0.51 (0.24) | 0.38 |
| | +SAF | 0.41 (0.2) | 0.32 (0.19) | 0.48 (0.2) | 0.47 (0.17) | 0.46 (0.2) | 0.57 (0.24) | 0.45 |
| | +CONTR. | 0.47 (0.23) | 0.3 (0.14) | 0.45 (0.24) | 0.32 (0.21) | 0.42 (0.22) | 0.44 (0.21) | 0.40 |
| 5K | PEBBLE | 0.17 (0.099) | 0.17 (0.089) | 0.28 (0.19) | 0.24 (0.15) | 0.23 (0.13) | 0.22 (0.12) | 0.22 |
| | +SAF | 0.36 (0.15) | 0.2 (0.13) | 0.4 (0.23) | 0.38 (0.17) | 0.19 (0.11) | 0.41 (0.2) | 0.32 |
| | +CONTR. | 0.34 (0.14) | 0.23 (0.19) | 0.52 (0.24) | 0.37 (0.2) | 0.4 (0.24) | 0.44 (0.18) | 0.38 |
| 2.5K | PEBBLE | 0.15 (0.086) | 0.13 (0.076) | 0.16 (0.1) | 0.16 (0.09) | 0.18 (0.075) | 0.25 (0.11) | 0.17 |
| | +SAF | 0.33 (0.19) | 0.12 (0.082) | 0.32 (0.17) | 0.18 (0.09) | 0.27 (0.11) | 0.22 (0.14) | 0.25 |
| | +CONTR. | 0.21 (0.13) | 0.19 (0.22) | 0.29 (0.17) | 0.17 (0.09) | 0.25 (0.15) | 0.28 (0.16) | 0.23 |

# G  Image Aug. Task Details

We present results ablating the impact of environment dynamics on top of the PEBBLE model to show how much of the REED gains come from encoding environment dynamics versus incorporating an auxiliary task. In our ablation, we replace the REED auxiliary task with an image-augmentation-based self-supervised learning auxiliary task that compares a batch of image observation states with augmented versions of the same observations using either LSS (Equation 3), or LC (Equation 4). We compare the impact of the SSL data augmentation auxiliary task (PEBBLE + Img. Aug.) with the impact of PEBBLE + REED on the image-based PEBBLE preference learning algorithm using the walker-walk, quadruped-walk, and cheetah-run DMC tasks. Policy and reward function learning is evaluated across feedback amounts and with the oracle labeler. All hyper-parameters match those specified in Appendix E, with the exception of those listed below (Table 8).

The Img. Aug. task learns representations of the state, not state-action pairs, as is done in REED, and so the Img. Aug. representations do not encode environment dynamics. To separate out states and actions, the Img. Aug. task uses the SAF reward model architecture. The data augmentations match those used in [58].

The PEBBLE + Img. Aug. algorithm is the same as PEBBLE + REED (Algorithm C), except, instead of updating the SPR network using the temporal dynamics task, the SSL Image Augmentation network (Appendix G.2) is updated using the image-augmentation task. The state encoder is shared between the reward and the SSL Image Augmentation networks.

The inclusion of an auxiliary task to improve the state encodings does improve the performance of PEBBLE, but does not improve as much as with the encoded environment dynamics. This can be seen across feedback types and DMC environments (see Figure 6 and Table 12). The performance improvement against the PEBBLE baseline suggests that having the auxiliary task does have some benefit. However, the performance improvements of REED against the image augmentation auxiliary task suggest that the performance improvements observed when the auxiliary task encodes environment dynamics gives meaningful improvements.

## G.1  Data Augmentation Parameters

This section presents the PEBBLE + Img. Aug. method for creating the augmented image observations. We evaluate the impact of both the "weak" and "strong" image augmentations used in [58]. The augmentation parameters for both the weak and strong styles are given in Table 9.

Table 8: Image Augmentation hyper-parameters for the PEBBLE reward + SSL model that differ from the parameters outlined in Appendix E.

| HYPER-PARAMETER | VALUE |
|---|---|
| REWARD MODEL | |
| Learning Rate | 1e-4 |
| Grayscale Images | False |
| Normalize Images | True |
| SSL DATA AUGMENTATION MODEL | |
| Learning Rate | 5e-5 |
| Grayscale Images | False |
| Normalize Images | True |
| Use Strong Augmentations (Table 9) | False |
| Batch Size | 256 |
| Loss | Distillation |

Table 9: Data Augmentation hyper-parameters for PEBBLE+SSL as specified in MOSAIC [58].

| HYPER-PARAMETER | VALUE |
|---|---|
| WEAK AUGMENTATIONS | |
| Random Jitter ($\rho$) | 0.01 |
| Normalization ($\mu, \sigma$) | [0.485, 0.456, 0.406], [0.229, 0.224, 0.225] |
| Random Resize Crop (scale min/max, ratio) | [0.7, 1.0], [1.8, 1.8] |
| STRONG AUGMENTATIONS | |
| Random Jitter ($\rho$) | 0.01 |
| Random Grayscale ($\rho$) | 0.01 |
| Random Horizontal Flip ($\rho$) | 0.01 |
| Normalization ($\mu, \sigma$) | [0.485, 0.456, 0.406], [0.229, 0.224, 0.225] |
| Random Gaussian Blur ($\sigma$ min/max, $\rho$) | [0.1, 2.0], 0.01 |
| Random Resize Crop (scale min/max, ratio) | [0.6, 1.0], [1.8, 1.8] |

## G.2 SSL Data Augmentation Architecture

The SSL Image Augmentation network is implemented in PyTorch. The architecture for the consistency_predictor comes from [58]. The image encoder architecture comes from [59]. The SSL network is initialized as follows:

```python
def build_ssl_network(self,
                      state_size: t.List[int],
                      state_embed_size: int,
                      consistency_projection_size: int,
                      consistency_comparison_hidden_size: int,
                      with_consistency_prediction_head: bool,
                      image_hidden_num_channels: int):
    """
    The network architecture and build logic to the complete the REED
    self-supervised temporal consistency task based on SPR.
    Args:
    state_size: dimensionality of the states
    state_embed_size: number of dimensions in the state embedding
    consistency_projection_size: number of units used to compare the
                                 predicted and target latent next
                                 state
    consistency_comparison_hidden_size: number of units in the hidden
                                        layers of the next_state_projector
                                        and the consistency_predictor
    with_consistency_prediction_head: when using the contrastive of
                                      objective the consistency
                                      prediction is not used, but is
                                      when using the distillation
                                      objective
    image_hidden_num_channels: the number of channels to use in the
                               image encoder's hidden layers
    """

    # Build the network that will encode the state features.
    state_conv_encoder = nn.Sequential(
        nn.Conv2d(state_size[0], image_hidden_num_channels, 3, stride=1),
        nn.ReLU(),
        nn.MaxPool2d(2, 2),
        nn.Conv2d(
            image_hidden_num_channels,
            image_hidden_num_channels,
            3, stride=1),
        nn.ReLU(),
        nn.MaxPool2d(2, 2),
        nn.Conv2d(
            image_hidden_num_channels,
            image_hidden_num_channels,
            3, stride=1),
        nn.ReLU(),
        nn.MaxPool2d(2, 2)
    )
    conv_out_size = torch.flatten(state_conv_encoder(torch.rand(size=[1] + list(state_size
    self.state_encoder = nn.Sequential(
        state_conv_encoder,
        nn.Linear(conv_out_size, state_embed_size),
        nn.LeakyReLU(negative_slope=1e-2)
    )

    self.consistency_projector = nn.Sequential(
        # Rearrange('B T d H W -> (B T) d H W'),
        nn.BatchNorm1d(state_embed_size), nn.ReLU(inplace=True),
        # Rearrange('BT d H W -> BT (d H W)'),
        nn.Linear(state_embed_size, consistency_comparison_hidden_size), nn.ReLU(inplace=
```

```
        nn.Linear(consistency_comparison_hidden_size, consistency_projection_size),
        nn.LayerNorm(consistency_projection_size)
)
# from: https://github.com/rll-research/mosaic/blob/561814b40d33f853aeb93f1113a301508
if with_consistency_prediction_head:
    self.consistency_predictor = nn.Sequential(
        nn.ReLU(inplace=True),
        nn.Linear(consistency_projection_size, consistency_comparison_hidden_size),
        nn.ReLU(inplace=True),
        nn.Linear(consistency_comparison_hidden_size, consistency_projection_size),
        nn.LayerNorm(consistency_projection_size))
else:
    self.consistency_predictor = None
```

A forward pass through the SSL network is as follows:

```
def ssl_forward(self,
                observations: RawAugmentedObservationsBatch,
                with_consistency_prediction_head: bool):
    """
    The logic for a forward pass through the SSL network.
    Args:
      observations: a batch of environment raw observations and
                    augmented observations
      with_consistency_prediction_head: when using the contrastive
                                        objective the consistency
                                        prediction is not used, but is
                                        when using the distillation
                                        objective
    Returns:
      predicted embedding of the augmented state and the augmented state embedding (detach
      dimensionality: (batch, time step)
    """

    # Encode the observations.
    observations_embed = self.state_encoder(observations.states)

    # Predict the augmented observations.
    if with_consistency_prediction_head:
        augmented_observation_pred = self.consistency_predictor(self.state_projector(obser
    else:
        augmented_observation_pred = self.state_projector(observations_embed)

    # we don't want gradients to back-propagate into the learned parameters from anything
    with torch.no_grad():
        # embed the augmented observation
        augmented_observation_embed = self.state_encoder(observations.augmented_states)
        # project the augmented observation embedding into a space where it can be compare
        projected_agumented_observation_embed = self.state_projector(augmented_observation

    return augmented_observation_pred, projected_agumented_observation_embed
```

## G.3    Image Aug. Auxiliary Task Results

We present results comparing the impact of REED's temporal auxiliary task and the Image Aug. auxiliary task. Learning curves (Figure 6) and normalized returns (Table 10) are provided for image-space observation walker-walk, cheetah-run, and quadruped-walk tasks across different amounts of feedback. We compare the contributions of the Image Aug. auxiliary task to PEBBLE against SAC trained on the ground truth reward, PEBBLE, PEBBLE + Distillation REED (+Dist.), and PEBBLE + Contrastive REED (+Contr.). Results are reported for the Image Aug. task using the distillation objective (Equation 3) as +Dist.+Img. Aug.

The inclusion of the Image Aug. auxiliary task improves performance relative to PEBBLE, but does not reach the level of performance achieved by REED. The gap policy performance between REED and the Image Aug. auxiliary task suggests that encoding environment dynamics in the reward function and not including an auxiliary task that trains on all policy experiences is the cause of the performance gains observed from REED.

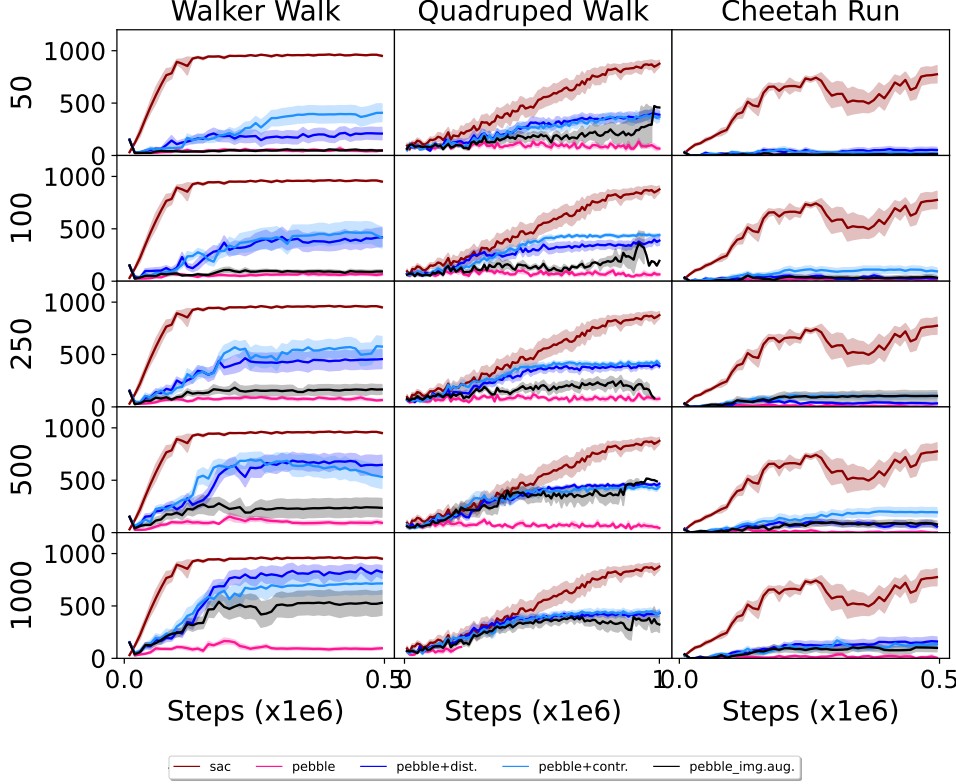

Figure 6: Episode returns learning curves for **walker walk**, **quadruped walk**, and **cheetah run** across preference-based RL methods and feedback amounts for image-based observations. The oracle labeller is used to generate preference feedback. Mean policy returns are plotted along the y-axis with number of steps (in units of 1000) along the x-axis. There is one plot per environment (grid columns) and feedback amount (grid rows) with corresponding results per learning methods in each plot. The learning methods evaluated are SAC trained on the ground truth reward, PEBBLE, PEBBLE with the Image Aug. auxiliary task (pebble+img. aug.), PEBBLE + Distillation REED (pebble+dist.), and PEBBLE + Contrastive REED (pebble+contr.). From top to bottom, the rows correspond to 2.5k, 5k, 10k, and 20k pieces of teacher feedback. From left to right, the columns correspond to walker walk, quadruped walk, and cheetah run.

Table 10: Ratio of policy performance on learned versus ground truth rewards for **walker-walk**, **cheetah-run**, and **quadruped-walk** across feedback amounts (with disagreement sampling). The results are reported as means (standard deviations) over 10 random seeds.

| | | DMC | | | | | |
|---|---|---|---|---|---|---|---|
| TASK | METHOD | 50 | 100 | 250 | 500 | 1000 | MEAN |
| WALKER-WALK | PEBBLE | 0.06 (0.02) | 0.07 (0.02) | 0.09 (0.02) | 0.11 (0.03) | 0.11 (0.03) | 0.09 |
| | +DIST. | 0.18 (0.03) | 0.33 (0.10) | 0.40 (0.09) | 0.57 (0.16) | **0.68 (0.23)** | 0.43 |
| | +CONTR. | **0.28 (0.12)** | **0.35 (0.13)** | **0.46 (0.14)** | **0.58 (0.14)** | 0.61 (0.16) | **0.46** |
| | +DIST.+IMG.AUG. | 0.06 (0.02) | 0.10 (0.02) | 0.16 (0.02) | 0.24 (0.03) | 0.46 (0.11) | 0.20 |
| QUADRUPED-WALK | PEBBLE | 0.28 (0.23) | 0.23 (0.19) | 0.23 (0.15) | 0.23 (0.19) | 0.47 (0.09) | 0.29 |
| | +DIST. | **0.56 (0.15)** | 0.59 (0.16) | 0.61 (0.12) | 0.71 (0.14) | **0.72 (0.19)** | 0.64 |
| | +CONTR. | 0.53 (0.16) | **0.64 (0.10)** | **0.69 (0.17)** | **0.73 (0.22)** | 0.71 (0.16) | **0.66** |
| | +DIST.+IMG.AUG. | 0.43 (0.19) | 0.35 (0.17) | 0.42 (0.17) | 0.68 (0.17) | 0.62 (0.16) | 0.50 |
| CHEETAH-RUN | PEBBLE | 0.01 (0.01) | 0.02 (0.01) | 0.02 (0.02) | 0.02 (0.03) | 0.03 (0.02) | 0.02 |
| | +DIST. | **0.06 (0.03)** | 0.07 (0.04) | 0.07 (0.02) | 0.12 (0.04) | **0.18 (0.07)** | 0.10 |
| | +CONTR. | 0.05 (0.02) | **0.14 (0.04)** | **0.14 (0.04)** | **0.23 (0.09)** | 0.16 (0.06) | **0.14** |
| | +DIST.+IMG.AUG. | 0.02 (0.01) | 0.05 (0.02) | **0.14 (0.05)** | 0.11 (0.04) | 0.12 (0.05) | 0.09 |

# H Complete Joint Results

Results are presented for all tasks, feedback amounts, and teacher labelling styles. The benefits of the SPR rewards are greatest: 1) for increasingly more challenging tasks, 2) when there is limited feedback available, and 3) when the labels are increasingly noisy.

## H.1 State-space Observations Learning Curves

Learning curves are provided for walker-walk (Figure 7), cheetah-run (Figure 8), quadruped-walk (Figure 9), button press (Figure 10), and sweep into (Figure 11) across feedback amounts and all teacher labelling strategies for state-space observations.

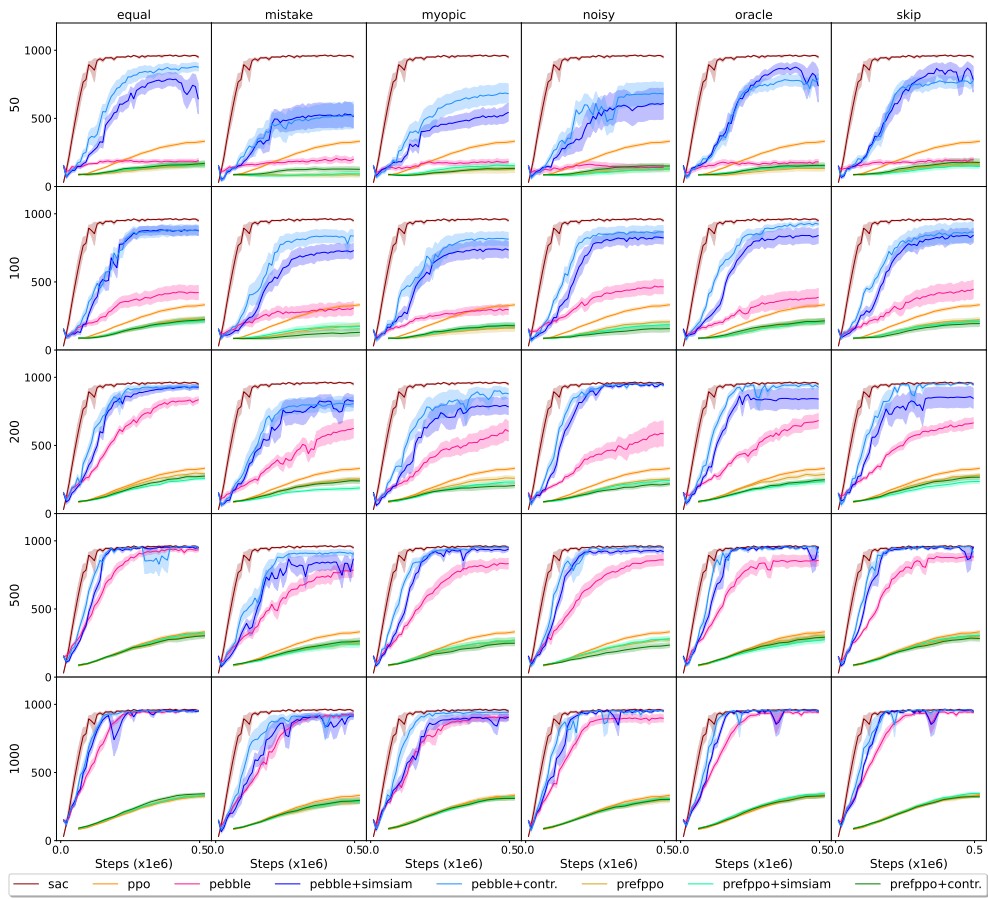

Figure 7: Returns learning curves for **walker-walk** across preference-based RL methods, labelling style, and feedback amount for state-space observations. Mean policy returns are plotted along the y-axis with number of steps (in units of 1000) along the x-axis. There is one plot per labelling style (grid columns) and feedback amount (grid rows) with corresponding results per learning methods in each plot. From top to bottom, the rows correspond to 50, 100, 500, and 1000 pieces of teacher feedback. From left to right, the columns correspond to equal, noisy, mistake, myopic, oracle, and skip labelling styles (see Appendix B for details).

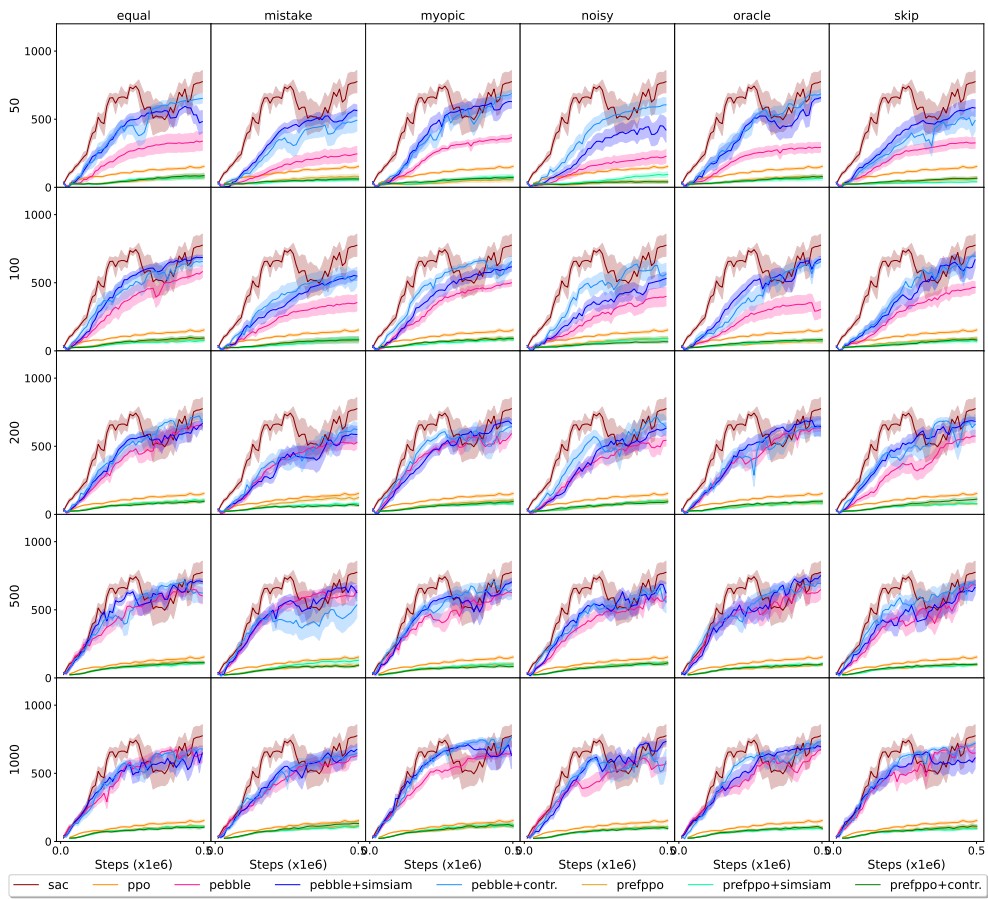

Figure 8: Returns learning curves for **cheetah-run** across preference-based RL methods, labelling style, and feedback amount for state-space observations. Mean policy returns are plotted along the y-axis with number of steps (in units of 1000) along the x-axis. There is one plot per labelling style (grid columns) and feedback amount (grid rows) with corresponding results per learning methods in each plot. From top to bottom, the rows correspond to 50, 100, 500, and 1000 pieces of teacher feedback. From left to right, the columns correspond to equal, noisy, mistake, myopic, oracle, and skip labelling styles (see Appendix B for details).

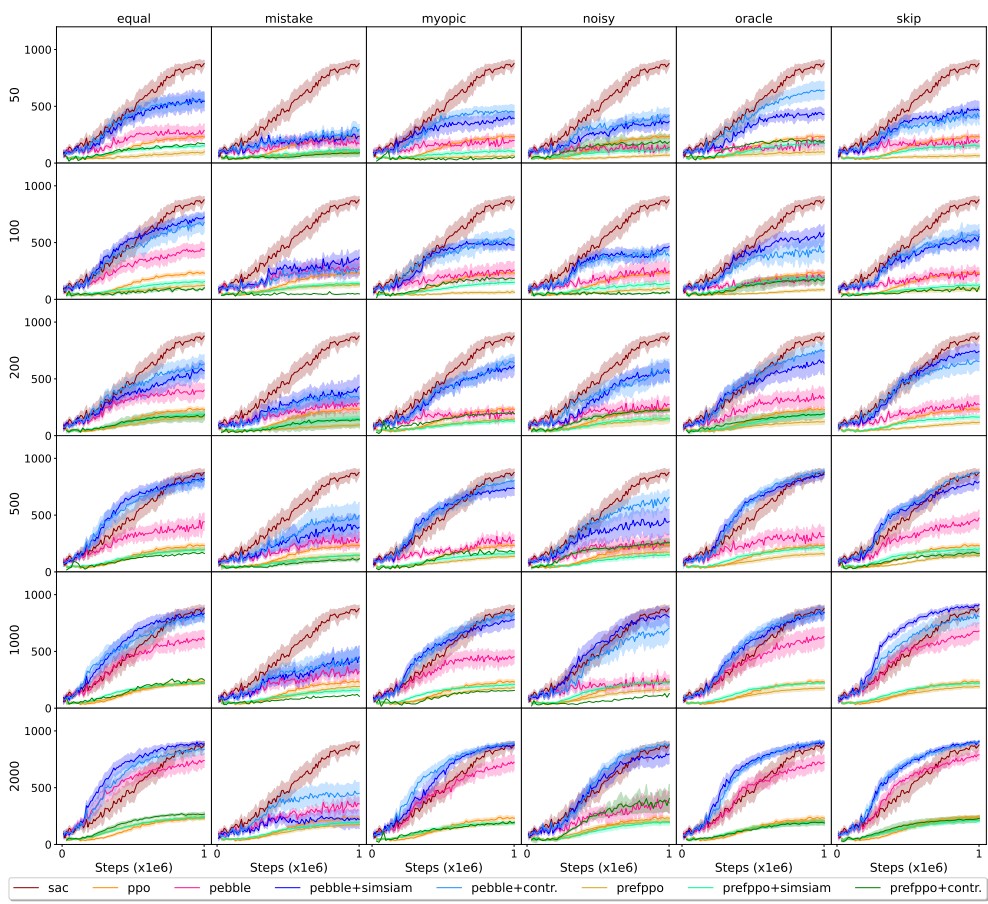

Figure 9: Returns learning curves for **quadruped-walk** across preference-based RL methods, labelling style, and feedback amount for state-space observations. Mean policy returns are plotted along the y-axis with number of steps (in units of 1e6) along the x-axis. There is one plot per labelling style (grid columns) and feedback amount (grid rows) with corresponding results per learning methods in each plot. From top to bottom, the rows correspond to 50, 100, 500, 1000, and 2000 pieces of teacher feedback. From left to right, the columns correspond to equal, noisy, mistake, myopic, oracle, and skip labelling styles (see Appendix B for details).

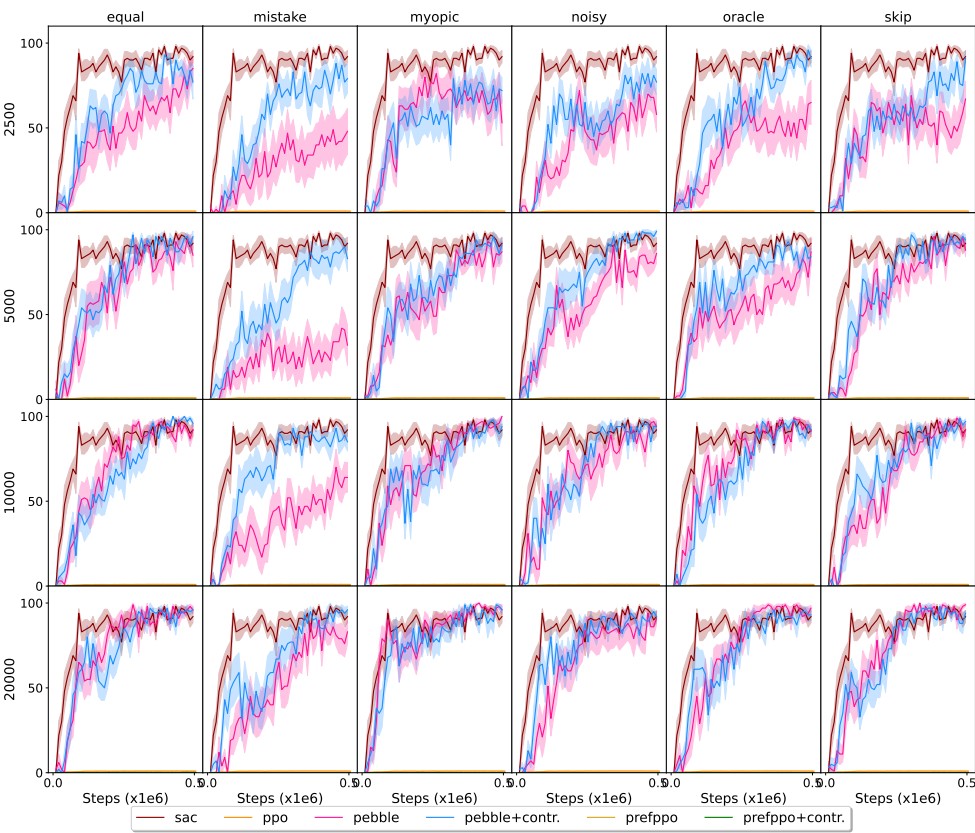

Figure 10: Success rate learning curves for **button press** across preference-based RL methods, labelling style, and feedback amount for state-space observations. Mean success rates are plotted along the y-axis with number of steps (in units of 1000) along the x-axis. There is one plot per labelling style (grid columns) and feedback amount (grid rows) with corresponding results per learning methods in each plot. From top to bottom, the rows correspond to 2.5k, 5k, 10k, and 20k pieces of teacher feedback. From left to right, the columns correspond to equal, noisy, mistake, myopic, oracle, and skip labelling styles (see Appendix B for details).

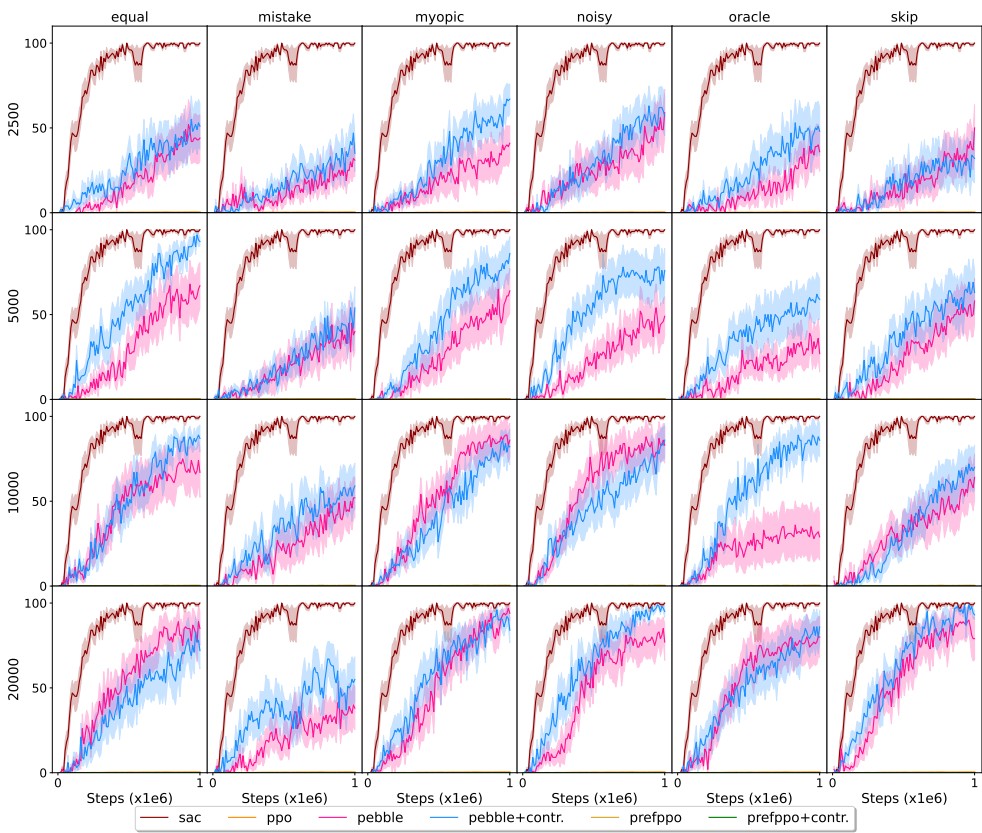

Figure 11: Success rate learning curves for **sweep into** across preference-based RL methods, labelling style, and feedback amount for state-space observations. Mean success rates are plotted along the y-axis with number of steps (in units of 1000) along the x-axis. There is one plot per labelling style (grid columns) and feedback amount (grid rows) with corresponding results per learning methods in each plot. From top to bottom, the rows correspond to 2.5k, 5k, 10k, and 20k pieces of teacher feedback. From left to right, the columns correspond to equal, noisy, mistake, myopic, oracle, and skip labelling styles (see Appendix B for details).

## H.2 State-space Normalized Returns

The normalized returns (see Section 6 and Equation 5) for walker-walk, quadruped-walk, cheetah-run, button press, and sweep into, across all teacher labelling styles and a larger range of feedback amounts for state-space observations, are given in Table 11.

Table 11: Ratio of policy performance on learned versus ground truth rewards for **walker-walk**, **cheetah-run**, **quadruped-walk**, **sweep into**, and **button press** across preference learning methods, labelling methods and feedback amounts (with disagreement sampling). The results are reported as means (standard deviations) over 10 random seeds.

| FEEDBACK | METHOD | ORACLE | MISTAKE | EQUAL | SKIP | MYOPIC | NOISY | MEAN |
|---|---|---|---|---|---|---|---|---|
| | | | | WALKER-WALK | | | | |
| | PEBBLE | 0.85 (0.17) | 0.76 (0.21) | 0.88 (0.16) | 0.85 (0.17) | 0.79 (0.18) | 0.81 (0.18) | 0.83 |
| | +DIST. | 0.9 (0.16) | 0.77 (0.2) | 0.91 (0.12) | 0.89 (0.16) | 0.8 (0.17) | 0.88 (0.17) | 0.86 |
| | +CONTR. | 0.9 (0.16) | 0.77 (0.2) | 0.91 (0.12) | 0.89 (0.16) | 0.8 (0.17) | 0.88 (0.17) | 0.86 |
| 1K | | | | | | | | |
| | PREFPPO | 1.0 (0.034) | 0.92 (0.056) | 1.0 (0.029) | 1.0 (0.031) | 1.0 (0.044) | 0.96 (0.056) | 0.99 |
| | +DIST. | 1.1 (0.025) | 0.92 (0.05) | 1.0 (0.031) | 1.0 (0.02) | 0.99 (0.035) | 0.96 (0.046) | 1.00 |
| | +CONTR. | 1.0 (0.029) | 0.93 (0.043) | 1.1 (0.021) | 1.0 (0.028) | 0.98 (0.033) | 0.95 (0.045) | 1.00 |
| | PEBBLE | 0.74 (0.18) | 0.61 (0.17) | 0.84 (0.19) | 0.75 (0.19) | 0.67 (0.19) | 0.69 (0.19) | 0.72 |
| | +DIST. | 0.86 (0.2) | 0.71 (0.2) | 0.87 (0.2) | 0.87 (0.2) | 0.82 (0.22) | 0.84 (0.2) | 0.83 |
| | +CONTR. | 0.9 (0.17) | 0.81 (0.19) | 0.9 (0.14) | 0.9 (0.17) | 0.88 (0.16) | 0.88 (0.18) | 0.88 |
| 500 | | | | | | | | |
| | PREFPPO | 0.95 (0.052) | 0.83 (0.087) | 0.95 (0.044) | 0.96 (0.058) | 0.89 (0.076) | 0.88 (0.069) | 0.91 |
| | +DIST. | 0.88 (0.074) | 0.81 (0.084) | 0.98 (0.025) | 0.95 (0.027) | 0.9 (0.073) | 0.9 (0.069) | 0.90 |
| | +CONTR. | 0.93 (0.061) | 0.85 (0.077) | 0.95 (0.046) | 0.92 (0.059) | 0.82 (0.085) | 0.77 (0.098) | 0.88 |
| | PEBBLE | 0.52 (0.17) | 0.46 (0.15) | 0.67 (0.2) | 0.54 (0.15) | 0.46 (0.13) | 0.45 (0.14) | 0.52 |
| | +DIST. | 0.74 (0.2) | 0.65 (0.21) | 0.79 (0.22) | 0.74 (0.19) | 0.64 (0.21) | 0.8 (0.25) | 0.73 |
| | +CONTR. | 0.84 (0.2) | 0.69 (0.2) | 0.84 (0.18) | 0.84 (0.2) | 0.75 (0.19) | 0.84 (0.21) | 0.80 |
| 200 | | | | | | | | |
| | PREFPPO | 0.93 (0.058) | 0.83 (0.076) | 0.93 (0.027) | 0.88 (0.045) | 0.87 (0.079) | 0.82 (0.06) | 0.88 |
| | +DIST. | 0.78 (0.087) | 0.68 (0.11) | 0.81 (0.062) | 0.77 (0.081) | 0.77 (0.077) | 0.78 (0.053) | 0.77 |
| | +CONTR. | 0.81 (0.088) | 0.81 (0.079) | 0.86 (0.044) | 0.85 (0.055) | 0.76 (0.13) | 0.74 (0.1) | 0.80 |
| | PEBBLE | 0.34 (0.11) | 0.31 (0.11) | 0.37 (0.1) | 0.37 (0.12) | 0.29 (0.085) | 0.41 (0.13) | 0.35 |
| | +DIST. | 0.68 (0.23) | 0.57 (0.2) | 0.72 (0.25) | 0.67 (0.23) | 0.61 (0.19) | 0.66 (0.24) | 0.65 |
| | +CONTR. | 0.78 (0.21) | 0.69 (0.22) | 0.74 (0.22) | 0.74 (0.19) | 0.68 (0.19) | 0.74 (0.22) | 0.73 |
| 100 | | | | | | | | |
| | PREFPPO | 0.68 (0.08) | 0.59 (0.093) | 0.73 (0.065) | 0.73 (0.065) | 0.58 (0.11) | 0.68 (0.072) | 0.67 |
| | +DIST. | 0.67 (0.08) | 0.63 (0.11) | 0.71 (0.075) | 0.71 (0.084) | 0.63 (0.099) | 0.63 (0.094) | 0.66 |
| | +CONTR. | 0.72 (0.084) | 0.49 (0.14) | 0.71 (0.063) | 0.65 (0.091) | 0.64 (0.1) | 0.58 (0.13) | 0.63 |
| | PEBBLE | 0.21 (0.1) | 0.22 (0.12) | 0.22 (0.12) | 0.23 (0.14) | 0.21 (0.11) | 0.18 (0.11) | 0.21 |
| | +DIST. | 0.66 (0.24) | 0.44 (0.13) | 0.6 (0.21) | 0.64 (0.24) | 0.44 (0.12) | 0.48 (0.15) | 0.54 |
| | +CONTR. | 0.62 (0.22) | 0.44 (0.11) | 0.72 (0.22) | 0.62 (0.22) | 0.54 (0.16) | 0.54 (0.17) | 0.58 |
| 50 | | | | | | | | |
| | PREFPPO | 0.51 (0.13) | 0.41 (0.18) | 0.57 (0.12) | 0.59 (0.11) | 0.51 (0.14) | 0.51 (0.14) | 0.52 |
| | +DIST. | 0.58 (0.13) | 0.41 (0.17) | 0.6 (0.13) | 0.56 (0.13) | 0.58 (0.12) | 0.48 (0.13) | 0.54 |
| | +CONTR. | 0.58 (0.12) | 0.54 (0.15) | 0.62 (0.12) | 0.63 (0.11) | 0.5 (0.13) | 0.57 (0.12) | 0.57 |

*Continues on next page...*

**TABLE 11** – *continued from previous page*

| FEEDBACK | METHOD | ORACLE | MISTAKE | EQUAL | SKIP | MYOPIC | NOISY | MEAN |
|---|---|---|---|---|---|---|---|---|
| | | | | CHEETAH-RUN | | | | |
| | PEBBLE | 0.87 (0.18) | 0.82 (0.18) | 0.91 (0.2) | 0.89 (0.17) | 0.87 (0.16) | 0.82 (0.18) | 0.86 |
| | +DIST. | 0.93 (0.2) | 0.83 (0.18) | 0.88 (0.14) | 0.92 (0.16) | 1.0 (0.18) | 0.86 (0.21) | 0.90 |
| | +CONTR. | 0.9 (0.17) | 0.84 (0.18) | 0.89 (0.17) | 0.96 (0.17) | 1.0 (0.21) | 0.85 (0.13) | 0.91 |
| 1K | | | | | | | | |
| | PREFPPO | 0.71 (0.064) | 0.72 (0.086) | 0.76 (0.069) | 0.71 (0.076) | 0.75 (0.066) | 0.69 (0.093) | 0.72 |
| | +DIST. | 0.67 (0.054) | 0.7 (0.076) | 0.75 (0.08) | 0.66 (0.064) | 0.79 (0.084) | 0.65 (0.081) | 0.70 |
| | +CONTR. | 0.7 (0.069) | 0.8 (0.11) | 0.72 (0.07) | 0.71 (0.065) | 0.78 (0.089) | 0.65 (0.083) | 0.73 |
| | PEBBLE | 0.86 (0.14) | 0.84 (0.18) | 0.86 (0.19) | 0.71 (0.16) | 0.79 (0.16) | 0.71 (0.15) | 0.79 |
| | +DIST. | 0.88 (0.22) | 0.83 (0.18) | 0.93 (0.15) | 0.76 (0.15) | 0.85 (0.14) | 0.8 (0.19) | 0.84 |
| | +CONTR. | 0.94 (0.21) | 0.72 (0.14) | 0.9 (0.21) | 0.89 (0.18) | 0.93 (0.18) | 0.82 (0.16) | 0.87 |
| 500 | | | | | | | | |
| | PREFPPO | 0.62 (0.043) | 0.63 (0.047) | 0.77 (0.089) | 0.66 (0.06) | 0.66 (0.04) | 0.72 (0.09) | 0.67 |
| | +DIST. | 0.67 (0.062) | 0.74 (0.14) | 0.7 (0.072) | 0.63 (0.069) | 0.67 (0.076) | 0.68 (0.081) | 0.68 |
| | +CONTR. | 0.66 (0.062) | 0.61 (0.072) | 0.73 (0.082) | 0.69 (0.065) | 0.61 (0.047) | 0.67 (0.073) | 0.66 |
| | PEBBLE | 0.71 (0.23) | 0.62 (0.22) | 0.71 (0.24) | 0.57 (0.2) | 0.75 (0.18) | 0.6 (0.22) | 0.66 |
| | +DIST. | 0.77 (0.28) | 0.61 (0.19) | 0.77 (0.22) | 0.79 (0.25) | 0.76 (0.25) | 0.65 (0.27) | 0.72 |
| | +CONTR. | 0.73 (0.22) | 0.67 (0.21) | 0.83 (0.25) | 0.8 (0.24) | 0.83 (0.19) | 0.76 (0.23) | 0.77 |
| 200 | | | | | | | | |
| | PREFPPO | 0.57 (0.042) | 0.73 (0.13) | 0.66 (0.059) | 0.54 (0.052) | 0.64 (0.073) | 0.56 (0.099) | 0.62 |
| | +DIST. | 0.53 (0.066) | 0.52 (0.047) | 0.66 (0.066) | 0.53 (0.049) | 0.55 (0.054) | 0.57 (0.085) | 0.56 |
| | +CONTR. | 0.62 (0.071) | 0.49 (0.046) | 0.61 (0.06) | 0.63 (0.1) | 0.56 (0.062) | 0.58 (0.048) | 0.58 |
| | PEBBLE | 0.4 (0.14) | 0.4 (0.13) | 0.61 (0.22) | 0.47 (0.2) | 0.55 (0.21) | 0.42 (0.14) | 0.48 |
| | +DIST. | 0.69 (0.26) | 0.59 (0.22) | 0.79 (0.28) | 0.65 (0.29) | 0.67 (0.26) | 0.53 (0.21) | 0.65 |
| | +CONTR. | 0.64 (0.28) | 0.65 (0.21) | 0.78 (0.21) | 0.7 (0.29) | 0.81 (0.26) | 0.72 (0.25) | 0.72 |
| 100 | | | | | | | | |
| | PREFPPO | 0.46 (0.036) | 0.51 (0.061) | 0.58 (0.051) | 0.58 (0.054) | 0.6 (0.047) | 0.59 (0.054) | 0.55 |
| | +DIST. | 0.49 (0.036) | 0.47 (0.088) | 0.49 (0.046) | 0.49 (0.038) | 0.52 (0.04) | 0.52 (0.081) | 0.50 |
| | +CONTR. | 0.54 (0.037) | 0.51 (0.06) | 0.59 (0.085) | 0.5 (0.059) | 0.59 (0.061) | 0.45 (0.039) | 0.53 |
| | PEBBLE | 0.35 (0.11) | 0.26 (0.098) | 0.39 (0.14) | 0.39 (0.12) | 0.4 (0.15) | 0.24 (0.089) | 0.34 |
| | +DIST. | 0.63 (0.23) | 0.59 (0.27) | 0.69 (0.25) | 0.62 (0.23) | 0.68 (0.31) | 0.46 (0.22) | 0.61 |
| | +CONTR. | 0.7 (0.28) | 0.51 (0.21) | 0.72 (0.28) | 0.53 (0.2) | 0.66 (0.28) | 0.66 (0.28) | 0.63 |
| 50 | | | | | | | | |
| | PREFPPO | 0.5 (0.066) | 0.49 (0.07) | 0.44 (0.076) | 0.4 (0.038) | 0.34 (0.062) | 0.3 (0.066) | 0.41 |
| | +DIST. | 0.44 (0.041) | 0.38 (0.039) | 0.44 (0.082) | 0.3 (0.063) | 0.44 (0.085) | 0.47 (0.11) | 0.41 |
| | +CONTR. | 0.47 (0.051) | 0.42 (0.039) | 0.47 (0.093) | 0.43 (0.038) | 0.46 (0.039) | 0.31 (0.061) | 0.43 |
| | | | | QUADRUPED-WALK | | | | |
| 2K | PEBBLE | 0.94 (0.15) | 0.55 (0.19) | 1.1 (0.26) | 1.0 (0.16) | 0.93 (0.13) | 0.56 (0.19) | 0.86 |
| | +DIST. | 1.3 (0.31) | 0.47 (0.19) | 1.4 (0.37) | 1.3 (0.26) | 1.2 (0.18) | 0.96 (0.15) | 1.09 |

*Continues on next page...*

| FEEDBACK | METHOD | ORACLE | MISTAKE | EQUAL | SKIP | MYOPIC | NOISY | MEAN |
|---|---|---|---|---|---|---|---|---|
| | +CONTR. | 1.3 (0.25) | 0.7 (0.16) | 1.2 (0.24) | 1.3 (0.29) | 1.3 (0.28) | 1.0 (0.16) | 1.13 |
| 2K | PREFPPO | 1.1 (0.18) | 0.89 (0.18) | 1.2 (0.22) | 1.2 (0.17) | 1.0 (0.25) | 1.1 (0.18) | 1.07 |
| | +DIST. | 1.1 (0.22) | 1.0 (0.2) | 1.2 (0.23) | 1.1 (0.24) | 1.1 (0.26) | 0.91 (0.1) | 1.06 |
| | +CONTR. | 1.0 (0.24) | 0.9 (0.2) | 1.4 (0.3) | 1.2 (0.29) | 1.1 (0.38) | 1.6 (0.32) | 1.28 |
| | PEBBLE | 0.86 (0.15) | 0.53 (0.19) | 0.88 (0.15) | 0.91 (0.14) | 0.73 (0.18) | 0.48 (0.25) | 0.73 |
| | +DIST. | 1.1 (0.19) | 0.59 (0.14) | 1.2 (0.22) | 1.3 (0.3) | 1.1 (0.21) | 1.0 (0.15) | 1.04 |
| | +CONTR. | 1.1 (0.19) | 0.63 (0.16) | 1.2 (0.29) | 1.1 (0.19) | 1.1 (0.19) | 0.83 (0.14) | 0.99 |
| 1K | PREFPPO | 0.9 (0.17) | 0.88 (0.17) | 1.1 (0.15) | 0.98 (0.21) | 0.89 (0.18) | 0.83 (0.17) | 0.92 |
| | +DIST. | 1.2 (0.21) | 0.88 (0.23) | 1.2 (0.27) | 1.2 (0.23) | 1.1 (0.26) | 1.1 (0.16) | 1.11 |
| | +CONTR. | 1.1 (0.19) | 0.68 (0.28) | 1.2 (0.25) | 1.1 (0.2) | 0.82 (0.31) | 0.56 (0.25) | 0.82 |
| | PEBBLE | 0.56 (0.21) | 0.48 (0.21) | 0.66 (0.2) | 0.64 (0.15) | 0.47 (0.22) | 0.48 (0.23) | 0.55 |
| | +DIST. | 1.1 (0.21) | 0.58 (0.16) | 1.2 (0.24) | 1.0 (0.22) | 1.0 (0.19) | 0.68 (0.16) | 0.93 |
| | +CONTR. | 1.1 (0.21) | 0.64 (0.11) | 1.1 (0.22) | 1.1 (0.17) | 1.0 (0.17) | 0.85 (0.14) | 0.97 |
| 500 | PREFPPO | 0.8 (0.18) | 0.81 (0.22) | 0.96 (0.12) | 0.72 (0.18) | 0.74 (0.24) | 0.88 (0.17) | 0.82 |
| | +DIST. | 1.1 (0.2) | 0.76 (0.19) | 1.0 (0.2) | 1.1 (0.25) | 0.89 (0.21) | 0.81 (0.25) | 0.95 |
| | +CONTR. | 1.1 (0.21) | 0.63 (0.25) | 0.9 (0.28) | 0.89 (0.22) | 0.88 (0.16) | 1.5 (0.48) | 0.95 |
| | PEBBLE | 0.54 (0.19) | 0.49 (0.22) | 0.64 (0.15) | 0.46 (0.2) | 0.43 (0.22) | 0.48 (0.23) | 0.51 |
| | +DIST. | 0.9 (0.17) | 0.57 (0.17) | 0.77 (0.16) | 0.89 (0.14) | 0.76 (0.11) | 0.68 (0.15) | 0.76 |
| | +CONTR. | 0.95 (0.15) | 0.53 (0.16) | 0.86 (0.16) | 0.88 (0.14) | 0.77 (0.12) | 0.74 (0.16) | 0.79 |
| 200 | PREFPPO | 0.7 (0.23) | 0.59 (0.28) | 0.82 (0.17) | 0.65 (0.27) | 0.8 (0.25) | 0.82 (0.27) | 0.73 |
| | +DIST. | 0.89 (0.15) | 0.79 (0.23) | 0.95 (0.18) | 0.87 (0.16) | 0.76 (0.26) | 0.79 (0.17) | 0.84 |
| | +CONTR. | 1.0 (0.33) | 0.7 (0.15) | 0.95 (0.21) | 0.86 (0.18) | 1.2 (0.43) | 1.2 (0.24) | 1.01 |
| | PEBBLE | 0.38 (0.21) | 0.47 (0.17) | 0.64 (0.14) | 0.42 (0.22) | 0.46 (0.2) | 0.44 (0.22) | 0.47 |
| | +DIST. | 0.78 (0.16) | 0.54 (0.2) | 0.98 (0.19) | 0.72 (0.15) | 0.75 (0.16) | 0.67 (0.18) | 0.74 |
| | +CONTR. | 0.67 (0.18) | 0.47 (0.2) | 0.89 (0.14) | 0.76 (0.15) | 0.79 (0.17) | 0.65 (0.19) | 0.71 |
| 100 | PREFPPO | 0.56 (0.31) | 0.81 (0.31) | 0.66 (0.22) | 0.62 (0.28) | 0.51 (0.31) | 0.6 (0.29) | 0.63 |
| | +DIST. | 1.0 (0.24) | 0.8 (0.23) | 0.82 (0.19) | 0.71 (0.24) | 0.76 (0.17) | 0.81 (0.26) | 0.82 |
| | +CONTR. | 0.91 (0.19) | 0.52 (0.42) | 0.61 (0.32) | 0.6 (0.27) | 0.99 (0.21) | 0.53 (0.37) | 0.69 |
| | PEBBLE | 0.38 (0.26) | 0.4 (0.23) | 0.49 (0.2) | 0.42 (0.25) | 0.42 (0.26) | 0.36 (0.26) | 0.41 |
| | +DIST. | 0.65 (0.16) | 0.47 (0.24) | 0.77 (0.14) | 0.68 (0.18) | 0.67 (0.2) | 0.56 (0.19) | 0.63 |
| | +CONTR. | 0.83 (0.12) | 0.49 (0.23) | 0.8 (0.14) | 0.65 (0.18) | 0.69 (0.16) | 0.62 (0.19) | 0.68 |
| 50 | PREFPPO | 0.68 (0.3) | 0.64 (0.28) | 0.58 (0.28) | 0.49 (0.26) | 0.49 (0.3) | 0.5 (0.31) | 0.56 |
| | +DIST. | 0.9 (0.19) | 0.71 (0.29) | 0.9 (0.25) | 0.83 (0.18) | 0.68 (0.26) | 0.77 (0.29) | 0.80 |
| | +CONTR. | 1.2 (0.34) | 0.58 (0.29) | 0.9 (0.27) | 0.82 (0.16) | 0.47 (0.44) | 1.1 (0.35) | 0.85 |

| FEEDBACK | METHOD | ORACLE | MISTAKE | EQUAL | SKIP | MYOPIC | NOISY | MEAN |
|---|---|---|---|---|---|---|---|---|
| | | | | BUTTON-PRESS | | | | |
| 20K | PEBBLE | 0.72 (0.26) | 0.57 (0.26) | 0.77 (0.25) | 0.75 (0.26) | 0.68 (0.21) | 0.72 (0.24) | 0.70 |
| | +CONTR. | 0.65 (0.25) | 0.61 (0.28) | 0.67 (0.27) | 0.67 (0.27) | 0.67 (0.24) | 0.69 (0.26) | 0.66 |
| | PREFPPO | 0.18 (0.03) | 0.18 (0.04) | 0.21 (0.03) | 0.18 (0.03) | 0.17 (0.04) | 0.17 (0.04) | 0.18 |
| | +CONTR. | 0.22 (0.03) | 0.17 (0.03) | 0.22 (0.02) | 0.17 (0.03) | 0.19 (0.03) | 0.17 (0.04) | 0.19 |
| 10K | PEBBLE | 0.66 (0.26) | 0.47 (0.21) | 0.67 (0.27) | 0.63 (0.26) | 0.67 (0.24) | 0.6 (0.26) | 0.62 |
| | +CONTR. | 0.65 (0.27) | 0.61 (0.3) | 0.66 (0.27) | 0.62 (0.26) | 0.6 (0.25) | 0.68 (0.28) | 0.64 |
| | PREFPPO | 0.18 (0.03) | 0.14 (0.04) | 0.19 (0.03) | 0.17 (0.04) | 0.18 (0.03) | 0.17 (0.04) | 0.17 |
| | +CONTR. | 0.15 (0.04) | 0.12 (0.05) | 0.18 (0.03) | 0.17 (0.03) | 0.17 (0.03) | 0.16 (0.03) | 0.16 |
| 5K | PEBBLE | 0.48 (0.21) | 0.31 (0.12) | 0.56 (0.25) | 0.54 (0.24) | 0.59 (0.23) | 0.52 (0.23) | 0.50 |
| | +CONTR. | 0.55 (0.24) | 0.54 (0.26) | 0.65 (0.27) | 0.63 (0.26) | 0.57 (0.24) | 0.63 (0.28) | 0.60 |
| | PREFPPO | 0.15 (0.04) | 0.13 (0.05) | 0.19 (0.03) | 0.16 (0.04) | 0.16 (0.04) | 0.14 (0.04) | 0.15 |
| | +CONTR. | 0.14 (0.04) | 0.13 (0.05) | 0.18 (0.03) | 0.14 (0.04) | 0.14 (0.04) | 0.14 (0.03) | 0.14 |
| 2.5K | PEBBLE | 0.37 (0.18) | 0.21 (0.088) | 0.44 (0.21) | 0.34 (0.15) | 0.4 (0.17) | 0.34 (0.18) | 0.35 |
| | +CONTR. | 0.49 (0.25) | 0.42 (0.22) | 0.52 (0.24) | 0.5 (0.23) | 0.44 (0.17) | 0.45 (0.21) | 0.47 |
| | PREFPPO | 0.14 (0.04) | 0.12 (0.05) | 0.13 (0.05) | 0.13 (0.05) | 0.13 (0.05) | 0.14 (0.05) | 0.13 |
| | +CONTR. | 0.14 (0.04) | 0.11 (0.05) | 0.15 (0.04) | 0.11 (0.04) | 0.14 (0.04) | 0.13 (0.04) | 0.13 |
| | | | | SWEEP-INTO | | | | |
| 20K | PEBBLE | 0.53 (0.25) | 0.26 (0.15) | 0.51 (0.23) | 0.52 (0.27) | 0.47 (0.28) | 0.47 (0.26) | 0.46 |
| | +CONTR. | 0.5 (0.22) | 0.36 (0.13) | 0.41 (0.2) | 0.6 (0.22) | 0.54 (0.21) | 0.61 (0.25) | 0.50 |
| | PREFPPO | 0.16 (0.046) | 0.14 (0.047) | 0.16 (0.069) | 0.18 (0.065) | 0.19 (0.063) | 0.08 (0.026) | 0.15 |
| | +CONTR. | 0.23 (0.064) | 0.11 (0.042) | 0.2 (0.058) | 0.2 (0.051) | 0.19 (0.054) | 0.1 (0.034) | 0.17 |
| 10K | PEBBLE | 0.28 (0.12) | 0.22 (0.13) | 0.45 (0.21) | 0.33 (0.17) | 0.47 (0.25) | 0.51 (0.24) | 0.38 |
| | +CONTR. | 0.47 (0.23) | 0.3 (0.14) | 0.45 (0.24) | 0.32 (0.21) | 0.42 (0.22) | 0.44 (0.21) | 0.40 |
| | PREFPPO | 0.16 (0.048) | 0.19 (0.064) | 0.18 (0.05) | 0.18 (0.049) | 0.12 (0.055) | 0.058 (0.024) | 0.15 |
| | +CONTR. | 0.11 (0.034) | 0.12 (0.046) | 0.15 (0.046) | 0.16 (0.056) | 0.11 (0.052) | 0.054 (0.029) | 0.12 |
| 5K | PEBBLE | 0.17 (0.099) | 0.17 (0.089) | 0.28 (0.19) | 0.24 (0.15) | 0.23 (0.13) | 0.22 (0.12) | 0.22 |
| | +CONTR. | 0.34 (0.14) | 0.23 (0.19) | 0.52 (0.24) | 0.37 (0.2) | 0.4 (0.24) | 0.44 (0.18) | 0.38 |
| | PREFPPO | 0.1 (0.039) | 0.078 (0.027) | 0.1 (0.032) | 0.092 (0.025) | 0.1 (0.038) | 0.051 (0.019) | 0.09 |
| | +CONTR. | 0.14 (0.052) | 0.097 (0.034) | 0.12 (0.032) | 0.14 (0.076) | 0.1 (0.026) | 0.043 (0.026) | 0.11 |
| 2.5K | PEBBLE | 0.15 (0.086) | 0.13 (0.076) | 0.16 (0.1) | 0.16 (0.088) | 0.18 (0.075) | 0.25 (0.11) | 0.17 |

| FEEDBACK | METHOD | ORACLE | MISTAKE | EQUAL | SKIP | MYOPIC | NOISY | MEAN |
|---|---|---|---|---|---|---|---|---|
| | +CONTR. | 0.21 (0.13) | 0.19 (0.22) | 0.29 (0.17) | 0.17 (0.092) | 0.25 (0.15) | 0.28 (0.16) | 0.23 |
| 2.5K | PREFPPO | 0.092 (0.032) | 0.097 (0.044) | 0.15 (0.051) | 0.15 (0.049) | 0.099 (0.035) | 0.032 (0.022) | 0.10 |
| | +CONTR. | 0.058 (0.019) | 0.048 (0.018) | 0.11 (0.032) | 0.072 (0.035) | 0.07 (0.016) | 0.036 (0.017) | 0.07 |

### H.3 Image-space Observations Learning Curves

Learning curves are provided for the walker-walk, cheetah-run, and quadruped-walk DMC tasks (Figure 12), and for the button press, sweep into, drawer open, drawer close, window open, and door close MetaWorld tasks (Figure 13) across feedback amounts for image-space observations. For all image-based results, the orable labeller is used to provide the preference feedback.

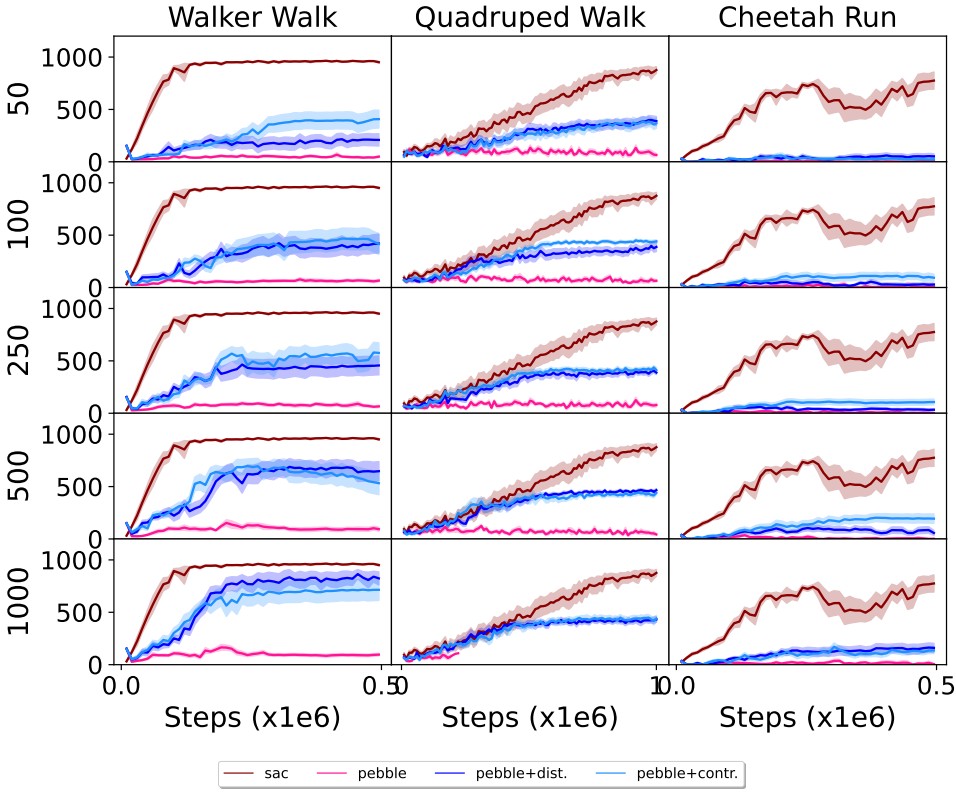

Figure 12: Episode returns learning curves for **walker walk**, **quadruped walk**, and **cheetah run** across preference-based RL methods and feedback amounts for image-based observations. The oracle labeller is used to generate preference feedback. Mean policy returns are plotted along the y-axis with number of steps (in units of 1000) along the x-axis. There is one plot per environment (grid columns) and feedback amount (grid rows) with corresponding results per learning methods in each plot. From top to bottom, the rows correspond to 2.5k, 5k, 10k, and 20k pieces of teacher feedback. From left to right, the columns correspond to walker walk, quadruped walk, and cheetah run.

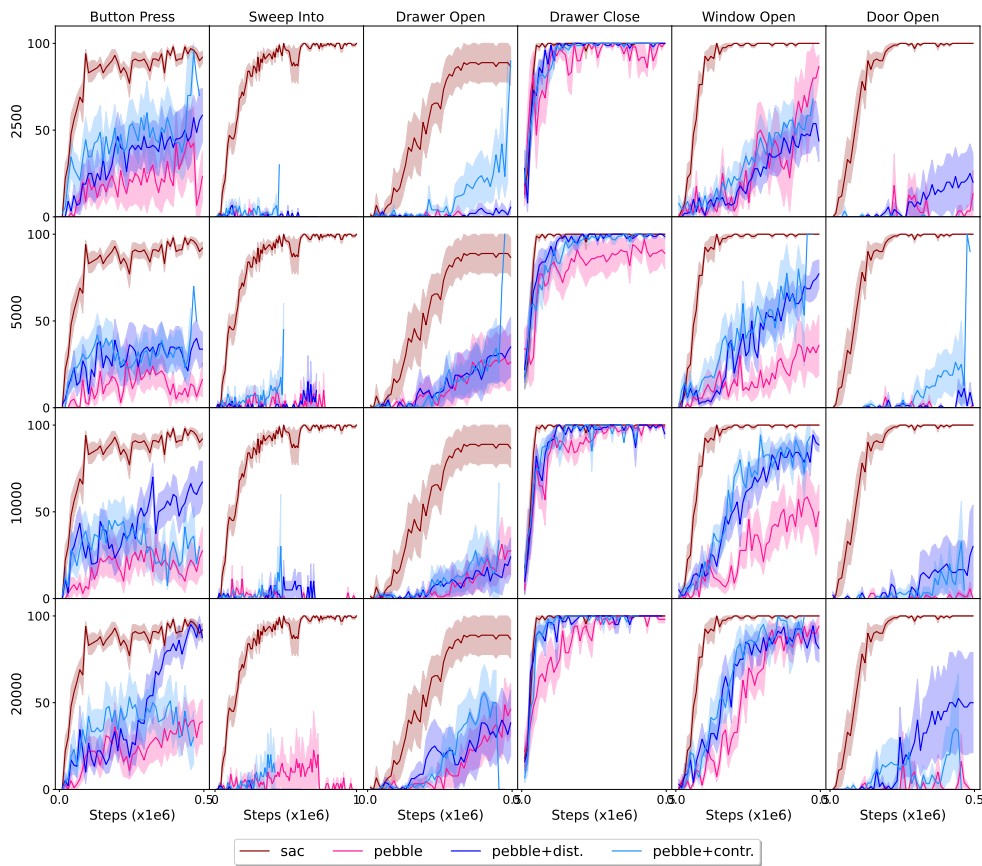

Figure 13: Success rate learning curves for **button press**, **sweep into**, **drawer open**, **drawer close**, **window open**, and **door close** across preference-based RL methods and feedback amounts for image-based observations. The oracle labeller is used to generate preference feedback. Mean success rates are plotted along the y-axis with number of steps (in units of 1000) along the x-axis. There is one plot per environment (grid columns) and feedback amount (grid rows) with corresponding results per learning methods in each plot. From top to bottom, the rows correspond to 2.5k, 5k, 10k, and 20k pieces of teacher feedback. From left to right, the columns correspond to button press, sweep into, drawer open, drawer close, window open, and door close.

## H.4 Image-space Observations Normalized Returns

The normalized returns (see Section **??** and Equation 5) for walker-walk, quadruped-walk, cheetah-run, button press, sweep into, drawer open, drawer close, window open, and door close across a larger range of feedback amounts for image-space observations are given in Table 12.

Table 12: Ratio of policy performance on learned versus ground truth rewards for the image-observations space **walker-walk**, **cheetah-run**, **quadruped-walk**, **sweep into**,**button press**, **drawer open**, **drawer close**, **window open**, and **door open** tasks across preference learning methods, labelling methods and feedback amounts (with disagreement sampling). The results are reported as means (standard deviations) over 10 random seeds.

| | | DMC | | | | | |
|---|---|---|---|---|---|---|---|
| TASK | METHOD | 50 | 100 | 250 | 500 | 1000 | MEAN |
| WALKER-WALK | PEBBLE | 0.06 (0.02) | 0.07 (0.02) | 0.09 (0.02) | 0.11 (0.03) | 0.11 (0.03) | 0.09 |
| | +DIST. | 0.18 (0.03) | 0.33 (0.10) | 0.40 (0.09) | 0.57 (0.16) | 0.68 (0.23) | 0.43 |
| | +CONTR. | 0.28 (0.12) | 0.35 (0.13) | 0.46 (0.14) | 0.58 (0.14) | 0.61 (0.16) | 0.46 |
| QUADRUPED-WALK | PEBBLE | 0.28 (0.23) | 0.23 (0.19) | 0.23 (0.15) | 0.23 (0.19) | 0.47 (0.09) | 0.29 |
| | +DIST. | 0.56 (0.15) | 0.59 (0.16) | 0.61 (0.12) | 0.71 (0.14) | 0.72 (0.19) | 0.64 |
| | +CONTR. | 0.53 (0.16) | 0.64 (0.097) | 0.69 (0.17) | 0.73 (0.22) | 0.71 (0.16) | 0.66 |
| CHEETAH-RUN | PEBBLE | 0.01 (0.01) | 0.02 (0.01) | 0.02 (0.02) | 0.02 (0.03) | 0.03 (0.02) | 0.02 |
| | +DIST. | 0.07 (0.03) | 0.07 (0.03) | 0.07 (0.02) | 0.12 (0.04) | 0.18 (0.07) | 0.10 |
| | +CONTR. | 0.05 (0.02) | 0.14 (0.04) | 0.14 (0.04) | 0.23 (0.09) | 0.16 (0.06) | 0.14 |

| | | METAWORLD | | | | MEAN |
|---|---|---|---|---|---|---|
| TASK | METHOD | 2.5K | 5K | 10K | 20K | |
| BUTTON PRESS | PEBBLE | 0.16 (0.07) | 0.20 (0.07) | 0.27 (0.09) | 0.33 (0.11) | 0.24 |
| | +DIST. | 0.25 (0.04) | 0.35 (0.09) | 0.46 (0.16) | 0.58 (0.23) | 0.41 |
| | +CONTR. | 0.27 (0.04) | 0.35 (0.10) | 0.34 (0.07) | 0.42 (0.11) | 0.35 |
| SWEEP INTO | PEBBLE | 0.03 (0.02) | 0.05 (0.03) | 0.04 (0.04) | 0.10 (0.08) | 0.06 |
| | +DIST. | 0.06 (0.05) | 0.10 (0.10) | 0.10 (0.06) | 0.10 (0.11) | 0.09 |
| | +CONTR. | 0.10 (0.11) | 0.12 (0.06) | 0.11 (0.07) | 0.12 (0.05) | 0.11 |
| DRAWER OPEN | PEBBLE | 0.39 (0.10) | 0.50 (0.12) | 0.55 (0.11) | 0.60 (0.14) | 0.51 |
| | +DIST. | 0.55 (0.07) | 0.60 (0.11) | 0.65 (0.08) | 0.70 (0.08) | 0.63 |
| | +CONTR. | 0.60 (0.11) | 0.60 (0.12) | 0.67 (0.08) | 0.71 (0.08) | 0.64 |
| DRAWER CLOSE | PEBBLE | 0.93 (0.22) | 0.82 (0.16) | 0.90 (0.17) | 0.86 (0.17) | 0.88 |
| | +DIST. | 0.95 (0.11) | 0.97 (0.12) | 0.93 (0.09) | 0.97 (0.06) | 0.96 |
| | +CONTR. | 0.97 (0.13) | 0.93 (0.11) | 0.93 (0.13) | 0.95 (0.08) | 0.95 |
| WINDOW OPEN | PEBBLE | 0.23 (0.16) | 0.18 (0.078) | 0.22 (0.1) | 0.35 (0.15) | 0.25 |
| | +DIST. | 0.26 (0.13) | 0.35 (0.17) | 0.49 (0.19) | 0.51 (0.19) | 0.40 |
| | +CONTR. | 0.26 (0.12) | 0.46 (0.20) | 0.48 (0.19) | 0.58 (0.19) | 0.44 |
| DOOR OPEN | PEBBLE | 0.17 (0.06) | 0.26 (0.06) | 0.23 (0.04) | 0.26 (0.06) | 0.23 |
| | +DIST. | 0.34 (0.07) | 0.33 (0.08) | 0.34 (0.09) | 0.49 (0.13) | 0.37 |
| | +CONTR. | 0.19 (0.03) | 0.34 (0.15) | 0.33 (0.07) | 0.40 (0.05) | 0.32 |

# I Stability and Generalization Benefits

## I.1 Reward Model Stability

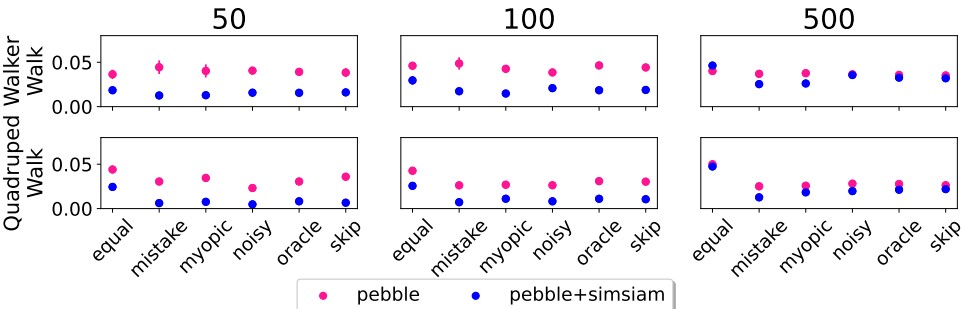

Figure 14: Variance in predicted $\hat{r}_\psi$ reward as $\hat{r}_\psi$ is learned and updated in conjunction with $\pi_\phi$. There is one plot per feedback amount (columns) and environment (rows) with corresponding results per learning method and labelling strategy in each plot. The learning methods assessed are PEBBLE and PEBBLE with the SimSiam temporal consistency objective. The labelling strategies (see Appendix B for details) are marked along the x-axis.

We evaluate reward model stability across updates by computing the variance in predicted rewards for each transition in $\mathcal{B}$ across reward model updates. We expect lower variance to translate into more stability and less reward non-stationarity, resulting in better policy performance. Mean and standard deviation in predicted rewards are provided for each model update over 10 random seeds in Figure 14. A representative subset of the conditions (walker-walk and quadruped-walk) from Section 6 are used to evaluate reward model stability.

Figure 14 shows that for fewer feedback samples, the predicted rewards are more stable across reward updates for REED methods. For larger amounts of feedback ($\geq 500$), where REED vs. non-REED reward policy performance is closer, the amount of predicted reward variability does not differ greatly between REED and non-REED reward functions. Therefore, the benefits of REED methods are most pronounced when preference feedback is limited.

These reward stability results partially explain why REED leads to better policy performance. Next, we investigate whether performance differences are solely due the interplay between reward and policy learning, or if the difference is also due to differences in overall reward quality.

## I.2 Reward Reuse

We assess reward re-usability on a representative subset of the conditions from Section 6 by: (1) learning a preference-based reward function following [3] and [4]; (2) freezing the reward function; (3) training a SAC policy from scratch using the frozen reward function. Reward function reuse is evaluated by comparing policy performance to: (a) SAC trained on the ground truth reward, and (b) SAC learned jointly with the reward function.

Figure 15 shows that, when reusing a reward function, REED improves policy performance relative to non-REED methods. When environment dynamics are encoded in the reward function, performance closely matches or exceeds that of both policies trained on the ground truth and policies trained jointly with the reward function.

We see different trends when comparing the reused case to the joint case across environments. For walker-walk, the policy trained on the reused reward function typically slightly under performs the policy trained in conjunction with the reward function (with the exception of feedback = 200 with the REED reward function) suggesting the REED and non-REED reward functions are over-fitting to the transitions they are trained on. For quadruped-walk, policies trained on the reused REED reward function outperform the policy trained jointly with the REED reward function for feedback amounts

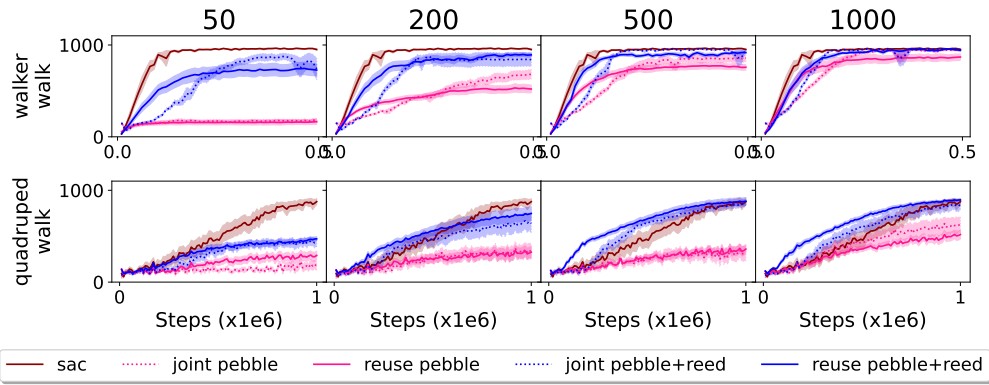

Figure 15: Preference-learned reward reuse policy learning curves for walker-walk and quadruped-walk comparing SAC on the ground truth reward function vs. on the PEBBLE learned reward function vs. on the PEBBLE with SimSiam reward function with the oracle labeller across feedback amount. Mean policy returns are plotted along the y-axis with number of steps along the x-axis.

$> 200$ and matches for 200. To our surprise, SAC is able to learn faster on the REED reward function than on the ground truth reward function whenever the amount of feedback is $> 200$. Whereas for the non-REED reward function, the policy trained on the reused reward function under-performs, matches, or out-performs the policy learned jointly with the reward function depending on the amount of feedback. The results suggest that the REED method are less prone to over-fitting.

### I.2.1 Complete Reward Reuse Results

The complete reward reuse results for walker-walk (Figure 16), quadruped-walk (Figure 17), and cheetah-run (Figure 18) across feedback amounts and teacher labelling strategies.

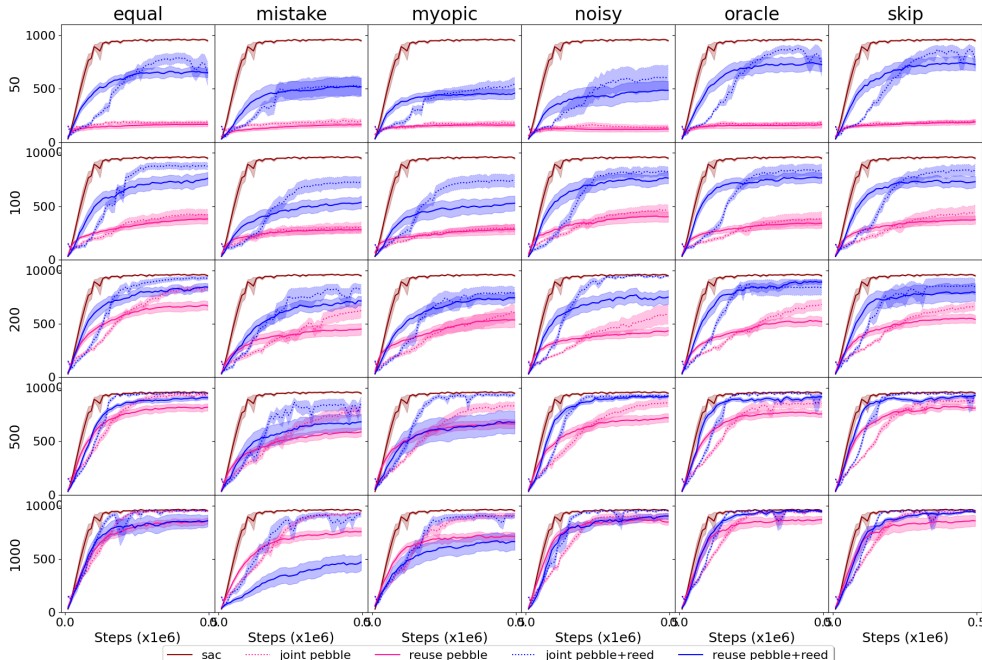

Figure 16: Learning curves for **walker-walk** comparing joint reward and policy learning with policy learning using a previously learned reward function across preference-based RL method, labelling style, and feedback amount. Mean policy returns are plotted along the y-axis with number of steps (in units of 1000) along the x-axis. There is one plot per labelling style (grid columns) and feedback amount (grid rows) with corresponding results per learning methods in each plot. From top to bottom, the rows correspond to 50, 100, 200, 500, and 1000 pieces of teacher feedback. From left to right, the columns correspond to equal, noisy, mistake, myopic, oracle, and skip labelling styles (see Appendix b for details).

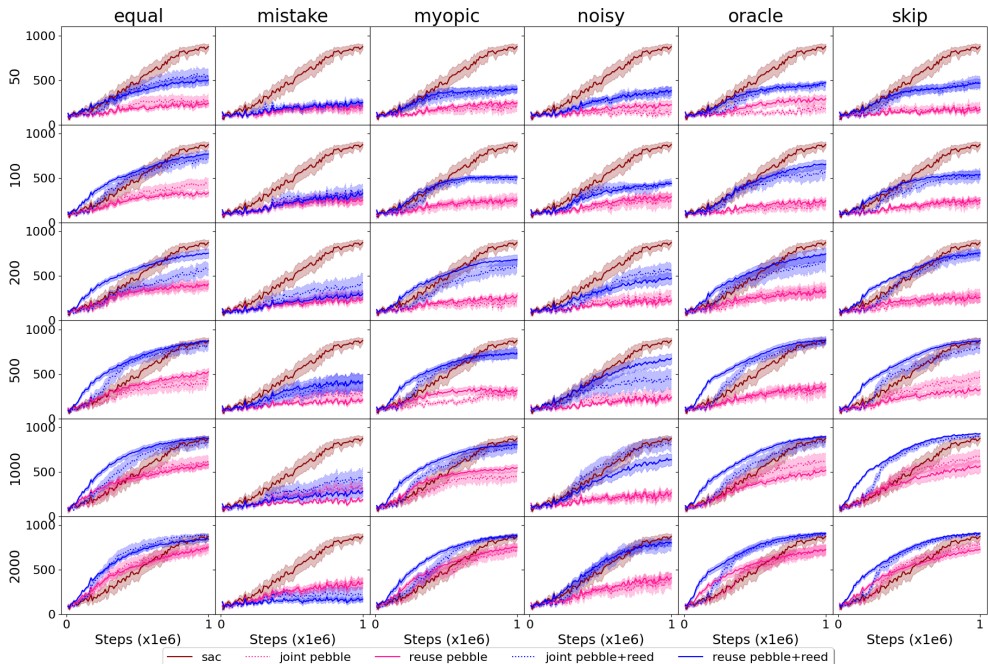

Figure 17: Learning curves for **quadruped-walk** comparing joint reward and policy learning with policy learning using a previously learned reward function across preference-based RL method, labelling style, and feedback amount. Mean policy returns are plotted along the y-axis with number of steps (in units of 1000) along the x-axis. There is one plot per labelling style (grid columns) and feedback amount (grid rows) with corresponding results per learning methods in each plot. From top to bottom, the rows correspond to 50, 100, 200, 500, 1000, and 2000 pieces of teacher feedback. From left to right, the columns correspond to equal, noisy, mistake, myopic, oracle, and skip labelling styles (see Appendix b for details).

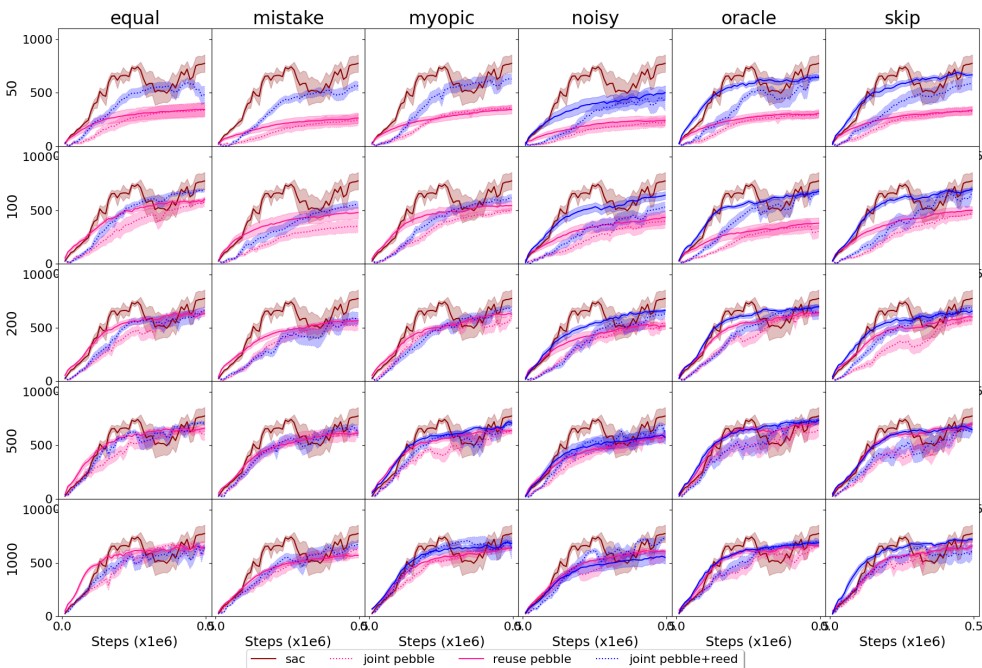

Figure 18: Learning curves for **cheetah-run** comparing joint reward and policy learning with policy learning using a previously learned reward function across preference-based RL method, labelling style, and feedback amount. Mean policy returns are plotted along the y-axis with number of steps (in units of 1000) along the x-axis. There is one plot per labelling style (grid columns) and feedback amount (grid rows) with corresponding results per learning methods in each plot. From top to bottom, the rows correspond to 50, 100, 200, 500, and 1000 pieces of teacher feedback. From left to right, the columns correspond to equal, noisy, mistake, myopic, oracle, and skip labelling styles (see Appendix b for details).

