# OpenReview forum: "Sample-Efficient Preference-based Reinforcement Learning with Dynamics Aware Rewards"
_robot-learning.org/CoRL/2023/Conference — CoRL 2023 Poster_

### Official Review · Reviewer_D6aA · 2023-07-12

**Confidence:** 3
**Originality:** Good
**Technical Quality:** Good
**Clarity Of Presentation:** Good
**Impact:** 2

**Recommendation:**

Weak Accept: I recommend accepting the paper, but will not argue for my recommendation if the majority of other reviewers have a different opinion.

**Review:**

The paper was clear for the most part and easy to read. While other works have also used auxillary and consistency losses for improvement of RL algorithms, I haven’t seen it used in the context of PbRL (except maybe a newish paper that proposes Model based PbRL). Additionally, this method performs an extensive comparison with other methods.

**Strengths**:
- comparison with several baselines
- study examining the advantage of dynamics encoding for noisy labels is especially interesting

**Weaknesses**:
- Comparison of other PEBBLE extensions (SURF, RUNE, MRN) for the labeling noise experiment was not performed.
- Possible typos and errors in the evaluation tables.
- The relevance of the dynamics  encoding especially in the context of sampling efficiency can be made clearer if the number of samples were plotted against the final reward for each of the compared methods (PEBBLE, SURF, RUNE, MRN, REED)

**Comments**:
- Since this method is learning a model anyway, instead of using SAC, has a model-based RL method investigated?
- instead of using an auxillary loss for dynamics learning, can $s_{t+1}$ be directly used as an input to the reward function?

**Quality Of The Limitations Section:**

Additional details required

**Questions For Rebuttal:**

1. Can the authors provide comparison of other PEBBLE extensions (SURF, RUNE, MRN) to REED for the labeling noise experiment?
2. Can the authors plot the final reward for each of the compared methods (PEBBLE, SURF, RUNE, MRN, REED) against the number of samples (feedbacks)? (It seems from the appendix that the DMC experiments were performed with six different amounts of feedback - but SURF, RUNE, MRN were not included in the evaluation)
3. Please address the following:
3-a. In line 138, "and then $\tilde y^d_{t+1}$ is pushed to be consistent with $z^s_{t+1}$ via a negative cosine similarity loss." $z^s_{t+1}$ should probably be $y^d_{t+1}$.
3-b. In Table 1., the SURF column seems to have larger mean values for the BUTTON PRESS and SWEEP examples, yet - +CONTRAST has been bolded.
3-c. In Figure 3, x- ticks on the left hand pictures look incorrect. - for example those for the cheetah run start with 10.0 and end at 0.5
3-d. incomplete plot for pebble in quadruped walk  (Figure 3 ) column 2. were all the steps not finished?


**Robotics Focus:**

Highly relevant to robotics but no hardware experiments

**Summary Of Paper:**

This paper introduced the notion of the encoding environment dynamics through a temporal consistency task during reward function learn. The authors claim that this inew method REED mproves the sample-efficiency of a preference-based reinforcement learning algorithm.
They compare their method with 3 other  reward-learning extensions to PEBBLE and 2 other PbRL algorithms (PEBBLE and PrefPPO) and find that in many cases their method outperforms the baselines. They additionally perform a set of studies to examine the effect of labeling errors,  feedback amounts etc. and compare the PbRL algorithms PEBBLE and PrefPPO against REED. They find that including the environment dynamics information improves the results.

**Summary Of Recommendation:**

I am familiar with works in the space of Preference based reinforcement learning and reinforcement learning in general

---

### Official Review · Reviewer_zDek · 2023-07-12

**Confidence:** 4
**Originality:** Good
**Technical Quality:** Good
**Clarity Of Presentation:** Good
**Impact:** 3

**Recommendation:**

Weak Accept: I recommend accepting the paper, but will not argue for my recommendation if the majority of other reviewers have a different opinion.

**Review:**

Strengths:
+The proposed method can take advantage of all experience by refining its latent state-action representation.
+Good range of experiments and ablations. The inclusion of learning curves and final normalized performance is nice.
+Good discussion and limitations sections.
+ Overall I like the idea of the paper and I think it covers an important problem.

Weaknesses:
- The authors say that giving feedback is difficult for single state-action pairs and cite TAMER; however, the TAMER papers shows that this is not hard and is a practical method for training RL systems.
-Missing some relevant prior work.
-Some of the design choices seem to introduce redundancy or are not adequately motivated.

**Quality Of The Limitations Section:**

Additional details required

**Questions For Rebuttal:**

Missing references:
Prior work has looked at learning latent representations for PbRL using a forward dynamics loss:
Brown et al. "Safe imitation learning via fast bayesian reward inference from preferences." ICML, 2020.

Prior work has looked at using contrastive learning to learn representations for PbRL:
Bobu et al. "SIRL: Similarity-based Implicit Representation Learning." HRI. 2023.

Why not use an exponential moving average like prior work BYOL and SPR?

Why use a linear projection followed by a linear prediction? This is redundant since two linear transformations can be represented by a single linear transformation.

On line 142 the term "different views" is not defined or explained.

Why first encode the states and actions separately?

Typo in Table 1: REED.SURF -- missing space

I'm not convinced that Table 1 shows a clear win for REED. SURF should be bolded for the BUTTON Press and SWEEP into tasks but is not. This means that REED methods are only significantly better on 5/10 tasks for PEBBLE and only on 4/10 tasks for PrefPPO. Especially in PrefPPO it seems that Base is the best method.

Why do we want to encode dynamics? Prior work has often only learned state-based reward functions. Indeed. AIRL argues that using actions will lead to learning the wrong reward function---one that is entangled with the dynamics function. How does your proposed approach perform when the dynamics change? It seems that REED would be especially sensitive to this as well as changes in embodiment. These issues should be discussed more in the limitations.

**Robotics Focus:**

Highly relevant to robotics but no hardware experiments

**Summary Of Paper:**

This paper seeks to improve preference-based reinforcement learning by learning the reward function based on a self-supervised representation that has been trained to predict dynamics. Results show that the proposed approach is often much more sample efficient than prior PbRL techniques.

**Summary Of Recommendation:**

Overall, I think the strengths outweigh the weaknesses so I would recommend a weak reject. However, there are no actual robotics experiments and some of the simulation experiments make me question the value of the proposed approach.

---
Post rebuttal: I thank the authors for their responses and maintain my score of weak accept.

---

### Official Review · Reviewer_XueR · 2023-07-19

**Confidence:** 4
**Originality:** Good
**Technical Quality:** Good
**Clarity Of Presentation:** Very Good
**Impact:** 3

**Recommendation:**

Weak Accept: I recommend accepting the paper, but will not argue for my recommendation if the majority of other reviewers have a different opinion.

**Review:**

Pros:
1. The paper provides extensive evaluation over preference learning benchmarks, as well as different types of labeling issues such as noisy, mistake and myopic.
2. The authors studies two self-supervised learning objectives to disentangle the overall improve from a specific objective choice, which better highlights the advantage of encoding environment dynamics in general.

Cons:
1. Missing stability and generalizability results. While the intro claims that leveraging environment dynamics improves the generalizability (contribution 2) and stability (contribution 3) of the learned reward functions, I was not able to find the evidences in the experiments. While the authors are able to show the results on better final performance of the learning method, these attributes needs to be separately validated.
2. More insight into better performance. While the authors are able to show that state-action representations learned with REED can yield better final performance, it remains unclear to me how this is portrayed in the benchmark environments. Does the learned reward function bypass certain limitations of previous methods? Can the authors provide some examples, or further interpretability study of the learned reward functions? While it is straightforward to understand that using environment dynamics SHOULD improve reward learning, more investigations can help reveal HOW.


**Quality Of The Limitations Section:**

Limitations are addressed clearly

**Questions For Rebuttal:**

Questions:
1. Can you provide evidence to support the claims that the reward functions learned with REED are more stable and generalizable?
2. Can you provide some concrete example to illustrate why the jointly-learned state-action representation can improve reward learning?







**Robotics Focus:**

Highly relevant to robotics but no hardware experiments

**Summary Of Paper:**

The authors using self-supervised objectives to learn a representation of the environment dynamics, which improves the result of preference-based learning. The learning objective can be plugged into existing reward learning frameworks such as PEBBLE. The results are competitive with other reward-learning techniques (especially when labeler is noisy) and highlight the importance of considering environment dynamics.


**Summary Of Recommendation:**

The author provides and straight-forward extension (learning joint state-action representations) to preference-based RL and show its advantage through extensive end-to-end experiments. While the proposed framework is solid, the paper could really benefit from further investigation and analysis of HOW the learned representations benefit reward learning. The authors can do this with concrete examples, or leverage interpretability tools.

---

### Official Review · Reviewer_8gBh · 2023-07-19

**Confidence:** 4
**Originality:** Fair
**Technical Quality:** Fair
**Clarity Of Presentation:** Good
**Impact:** 2

**Recommendation:**

Weak Accept: I recommend accepting the paper, but will not argue for my recommendation if the majority of other reviewers have a different opinion.

**Review:**

Strengths:
1. The authors examine a new architecture for predicting reward functions where the state representations are learned using SPR (self-predictive representations).
2. The authors perform extensive experiments demonstrating the superiority of their method in the 5 environments considering Varying preferences, and varying noise strategy in labelling preferences and adding their proposed modifications to an off-policy PbRL variant (Pebble) and on-policy (PrefPPO).

Weaknesses:
1. Lack of novelty: I am not sure as to why the proposed modification is novel. The idea of using representations predictive of environment dynamics for learning reward functions have been explored before. eg. (Sec 5.2 in https://arxiv.org/pdf/2002.09089.pdf). Basically, a literature review of a whole line of work in learning from preferences seems to be ignored:
a. https://arxiv.org/abs/1904.06387
b. https://arxiv.org/pdf/2002.09089.pdf
c. https://arxiv.org/abs/2202.03481
d. https://arxiv.org/abs/2010.11723

2. Experimental results: For a majority of tasks in Table 1, it seems that the improvement by REED falls within the standard deviation of results from other methods. Are the results statistically significant?

3. Rewards encoding environment dynamics: I am not sure how this claim can be made. Traditionally, in prior literature, the rewards that are shaped by environment dynamics are referred to as Potential based rewards (see https://people.eecs.berkeley.edu/~russell/papers/icml99-shaping.pdf). But, in this paper, only the representation of states encode environment dynamics, not rewards that are shaped by dynamics. Reward shaping has theoretical guarantee on learning performance but learning a state dependent on reward has no clear impact. Second, even the raw state directly encodes environment dynamics as it is a Markov decision process. I am not sure if the claims of the paper hinges on reward encoding environment dynamics. An alternate hypothesis for the performance gains might just be the regularization induced by the bottleneck of learning state features.

**Quality Of The Limitations Section:**

Limitations are addressed clearly

**Questions For Rebuttal:**

1. The reason for incremental improvement in PrefPPO is given to be the lack of diversity in replay buffer. Why not maintain another buffer that stores all previous transitions for the purpose of training state representations?

**Robotics Focus:**

Highly relevant to robotics but no hardware experiments

**Summary Of Paper:**

The paper proposes to learn reward functions using a learned representation of states that can predict the dynamics of the environment. Two kinds of objectives are explored for learning state-representation - Distillation and Contrastive. The reward is then constructed as a linear or a simple function of the learned representations. The authors compare their method on the B-pref benchmark suite and show improved performance.

**Summary Of Recommendation:**

I am having a hard time understanding the novelty of the paper. As I pointed out, similar ideas have already been executed in prior work but I appreciated the experimental evaluation of this paper.


Update: Thanks for the response, I have updated my score. The author's reply about updating the paper to reflect novelty clearly (that objective they proposed has been used before and contribution is the empirical evaluation between using dynamics aware representation loss vs non-dynamics aware representation loss for PbRL). I am still conflicted about the name of the method which is misleading.

---

### Author Response · Authors · 2023-08-10
**Polite follow up to reviewers**

If you have further questions, comments, or feedback given our rebuttals, please let us know so that we can do our best to address your points in the five days that are remaining.  Thank you for your time considering our paper.

---

### Decision · Program_Chairs · 2023-08-30

**Decision:**

Accept (Poster)

**Comment:**

The paper initially got mixed reviews with identified weaknesses in terms of novelty and significance of the results, and lack of ablation studies. Yet, these concerns have been addressed in the rebuttal and all reviewer see the merits of the presented approach. Yet, as suggestd by reviewer 8gBh the focus should be put more on the analysis of the results and less on the algorithmic novelty as similar predictive state respresentations have been already proposed. I agree with the reviewers that the paper contans interesting insights of how to construct state representations for preference-based RL and recommend acceptance.